# Linear and phase controllable terahertz frequency conversion via ultrafast breaking the bond of a meta-molecule

Siyu Duan[1,2], Xin Su[1,3], Hongsong Qiu[1], Yushun Jiang[1], Jingbo Wu [1,2] ✉, Kebin Fan[1,2], Caihong Zhang[1,2], Xiaoqing Jia [1,2], Guanghao Zhu[1], Lin Kang[1,2], Xinglong Wu[4,5], Huabing Wang [1,2], Keyu Xia [3,4,6] ✉, Biaobing Jin [1,2] ✉, Jian Chen [1,2] & Peiheng Wu [1,2]

The metasurface platform with time-varying characteristics has emerged as a promising avenue for exploring exotic physics associated with Floquet materials and for designing photonic devices like linear frequency converters. However, the limited availability of materials with ultrafast responses hinders their applications in the terahertz range. Here we present a time-varying metasurface comprising an array of superconductor-metal hybrid meta-molecules. Each meta-molecule consists of two meta-atoms that are "bonded" together by double superconducting microbridges. Through experimental investigations, we demonstrate high-efficiency linear terahertz frequency conversion by rapidly breaking the bond using a coherent ultrashort terahertz pump pulse. The frequency and relative phase of the converted wave exhibit strong dependence on the pump-probe delay, indicating phase controllable wave conversion. The dynamics of the meta-molecules during the frequency conversion process are comprehensively understood using a time-varying coupled mode model. This research not only opens up new possibilities for developing innovative terahertz sources but also provides opportunities for exploring topological dynamics and Floquet physics within metasurfaces.

In classical and quantum systems, the coupling of resonators exhibits numerous interesting phenomena, including non-Hermitian physics[1–3], electromagnetic-induced transparency[4–6], and bound states in the continuum[7–10]. These systems offer potential applications such as slowing down light[11–13], sensitive sensing[14–16], optical switching[17,18], and information storage[19]. When the switching speed of the coupling coefficient is comparable to the wave oscillation period, temporal modulation reveals rich physical phenomena[20], such as nonreciprocity[21,22], and photon acceleration[23–25]. In particular, two coupled resonators can form an artificial molecule[26,27], providing a platform for engineering molecules and exploring molecular dynamics with time-varying properties. However, the impact of time-dependent coupling has yet to be thoroughly studied due to the lack of suitable devices and fast modulation methods.

[1]Research Institute of Superconductor Electronics (RISE) & Key Laboratory of Optoelectronic Devices and Systems with Extreme Performances of MOE, School of Electronic Science and Engineering, Nanjing University, Nanjing 210023, China. [2]Purple Mountain Laboratories, Nanjing 211111, China. [3]College of Engineering and Applied Sciences, Nanjing University, Nanjing 210023, China. [4]National Laboratory of Solid State Microstructures, Nanjing University, Nanjing 210093, China. [5]School of Physics, Nanjing University, Nanjing 210093, China. [6]Shishan Laboratory, Suzhou Campus of Nanjing University, Suzhou 215000, China. ✉e-mail: jbwu@nju.edu.cn; keyu.xia@nju.edu.cn; bbjin@nju.edu.cn

In contrast to static metasurfaces, time-varying metasurfaces offer numerous opportunities for dynamically modulating electromagnetic waves[20,28]. They are extensively utilized to investigate fundamental physics, such as the inverse Doppler effect[29], and hold promise for exotic applications[30,31], including nonreciprocal transmission[32,33] and frequency conversion[34-36]. At microwave frequencies, the periodically time-modulated metasurfaces have exhibited application prospects in wireless communication, radar, and other fields[37,38]. Despite the rapid progress, developing the terahertz (THz) time-modulation devices still faces tremendous challenges since the modulation speed of electronic switches limits the extent to which the input signal frequency can be shifted.

By employing the photoconductive effect to achieve ultrafast carrier generation in the semiconductor, two small meta-atoms can merge into a larger one, thereby creating a time-varying boundary in the metasurface[39]. Linear THz frequency conversion was experimentally observed in such metasurface platforms[39,40]. It was theoretically proposed that phase control can be used for wavefront engineering of the converted waves. In recent work, efficient frequency conversion, as well as phase coherence of the converted waves, has been observed experimentally by ultrafast modulation of structural dispersion in the waveguide[41] or loss in the Fabry–Perot cavity[42]. Despite the progress, a chip-compatible planar frequency converter with a high conversion efficiency and phase control is highly desired to advance the practical applications.

Phase-controlled linear frequency conversion represents a promising approach for developing new THz sources, which are highly sought after in applications such as next-generation communication, imaging[43,44], and radio astronomy[45]. Currently, wave conversion based on microwave frequency multiplication or laser frequency beating is the primary mechanism for THz generation[46]. However, the conversion efficiency remains low, necessitating the use of high-power microwave sources or lasers.

In this study, we present a conceptual demonstration of linear and phase controllable THz frequency conversion with high efficiency by ultrafast "breaking" the bond of the superconducting-metal hybrid meta-molecule. Each meta-molecule consists of two metallic resonators, referred to as meta-atoms, coupled via two superconducting microbridges (Fig. 1a). The pair of superconducting microbridges serves as the molecular bond, and their sensitivity to the external THz pump is crucial for breaking the bond, as the THz pump can rapidly suppress superconductivity (Fig. 1b). The meta-molecules periodically arrange to form the metasurface. Unlike other metasurfaces, the bond in our system is time-dependent. Therefore, we develop a coupled mode model to analyze the spectral response with a time-varying coupling strength ($J$) (Fig. 1c). When the intense pump pulse transiently disrupts the bond, $J$ undergoes temporal variations. Consequently, the energy levels of the meta-molecule are modified, and the transition of electrons from excited states to the ground state results in the emission of new THz photons, enabling highly efficient frequency conversion. The measured conversion efficiency can reach up to 4.1 %. Furthermore, the conversion efficiency can be conveniently controlled by electrically tuning the superconducting microbridges. These electrically tunable meta-molecules with time-varying bonds represent a significant advancement toward THz coherent manipulation, as well as the exploration of molecular dynamics and topological physics on a metasurface platform.

## Results

### Design of time-varying meta-molecule

We designed a superconductor-metal hybrid meta-molecule array, as illustrated in Fig. 1a, b. Basically, the meta-molecule consists of two "bonded" meta-atoms, which are two mirror-symmetric planar metallic resonators connected via two superconducting microbridges. Details

on the geometric parameters, simulated transmission spectra, and fabrication processes are provided in Supplementary Note 1. External stimuli such as temperature, current, and THz pulses can transition the NbN microbridges from the superconducting to the normal state and break the bond correspondingly[47]. Compared to thermal and voltage methods, an intense THz pump pulse has an advantage because it can break the bond ultrafast within a few picoseconds[48,49]. However, in practice, when the NbN microbridge goes into the normal state, it becomes lossy, resulting in a weak bond between the two meta-atoms. As a result, the hybrid meta-molecule bond can be modified ultrafast in time in comparison with the oscillation period of a THz probe wave. The schematic diagram of the THz pump-probe spectroscopy measurement is shown in Fig. 1b. The arrival time of the pump and probe pulse peaks to the meta-molecule array is $t_J$ and $t_p$, respectively. The time delay is defined as $t_{pp} = t_p - t_J$. In experiments, we can only determine the rough arrival times $t_J$ and $t_p$ because the fields are imperfect.

### Time-varying coupled mode model

In the experiment, the meta-molecule is formed by two bonded and mirror-symmetric meta-atoms. Thus, we can simply consider that the two meta-atoms are driven by a THz probe pulse of $E_p(t)$ with the same phase and strength ($\kappa_e$). The two meta-atoms have the resonance frequencies of $f_a$ and $f_b$, and the total loss rates of $\kappa_a$ and $\kappa_b$. We use the annihilation operators of $a$ and $b$ for the two resonator modes. The meta-atoms are coupled via the microbridges with a strength of $J(t)$. A pump pulse is used to tune this coupling strength temporally. During the time-varying process, the superconductivity of the microbridges is suppressed rapidly, and they are switched to the normal state, leading to an increased Ohmic loss to the meta-atoms. We introduce a time-varying loss rate of $\kappa_J(t)$ to the meta-atoms for modeling the effect of varying coupling on the system loss.

The spectral response of the hybrid meta-molecule can be comprehended using a time-varying coupled mode model. The Hamiltonian for a typical system comprising two coupled meta-atoms can be expressed as[17,50]:

$$
\begin{aligned}
H = & f_a a^\dagger a + f_b b^\dagger b + [J(t)a^\dagger b + J^*(t)ab^\dagger] \\
& + i\sqrt{2\kappa_e}E_p(t)(a^\dagger + a) \\
& + i\sqrt{2\kappa_e}E_p(t)(b^\dagger + b),
\end{aligned} \tag{1}
$$

The first two terms describe the free energy of the meta-atoms. The third term indicates the time-varying coupling between two meta-atoms. The last two terms are for the driving of the two meta-atoms. The decay of the meta-atoms is included in the model using the Lindblad operators. As well known, the spectral response is highly sensitive to the bond or coupling strength. By incorporating a time-dependent bond into the meta-molecule, we can effectively model frequency conversion in the transition region. According to the Hamiltonian in Eq. 1 and the mechanism shown in Fig. 1c, the frequency shift ($\Delta f_c$) between the converted and the input field is roughly given by

$$
\Delta f_c = |J|_{max} - |J(t_J)| \tag{2}
$$

The phase shift between the two fields includes two contributions: (i) the phase difference when the supermode of the meta-molecule is excited; (ii) the phase accumulating during the time-dependent radiation when the coupling varies. Thus, it is difficult to provide an analytical estimation. The phase of the converted field will be experimentally measured and numerically calculated.

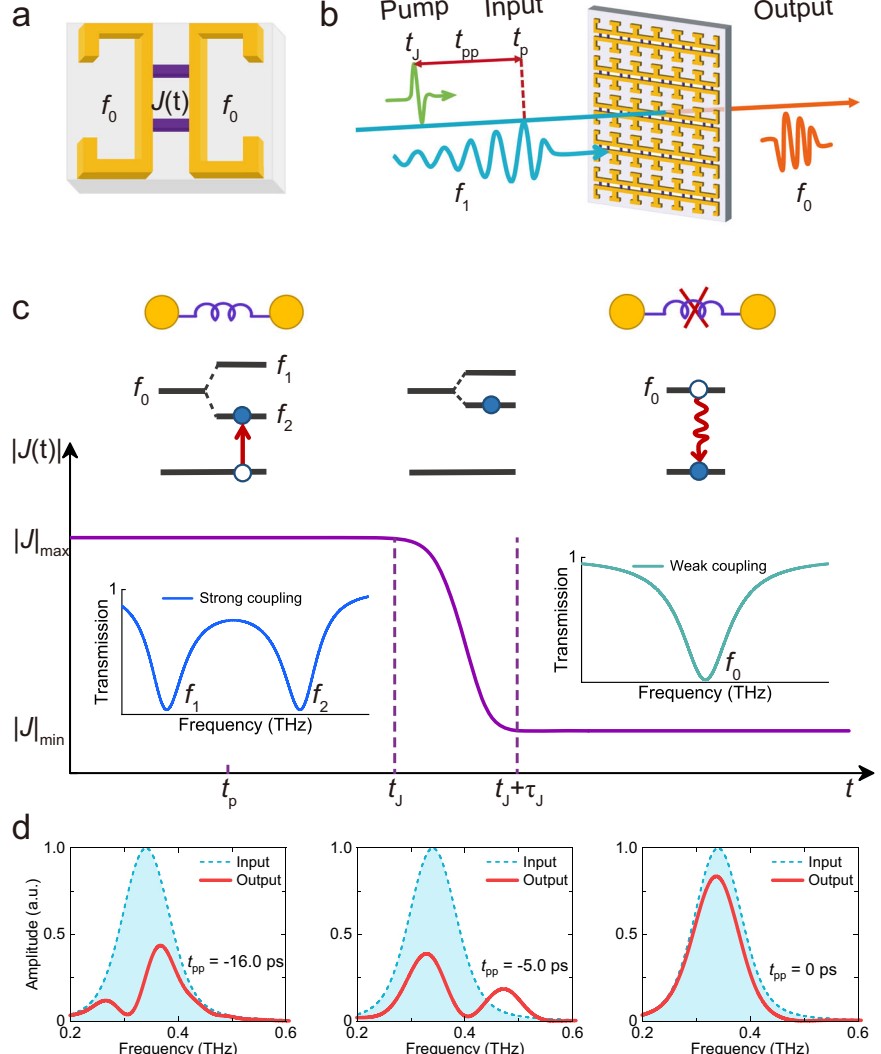

**Fig. 1 | Conceptual representation of frequency conversion in meta-molecules with a time-varying bond. a** Schematic of a meta-molecule composed of two coupled meta-atoms (metallic resonators). The time-varying bond strength of the molecule is denoted by $J(t)$, while $f_0$ represents the resonance frequency of the resonators. **b** Diagram illustrating the hybrid meta-molecule array and the experimental setup for THz pump-probe spectroscopy measurements. The arrival times of the pump and probe pulse peaks at the hybrid meta-molecule array are indicated as $t_J$ and $t_p$, respectively. The time delay is defined as $t_{pp} = t_p - t_J$. The input wave has a center frequency of $f_1$, and the frequency of the output wave shifts to $f_0$. **c** Absolute value of coupling coefficient $|J|$ as a function of time. The top panel displays the bonding state of meta-molecules and energy level diagrams at different time intervals. $\tau_J$ corresponds to the period required for breaking the meta-molecular bond. The calculated transmission spectra of the meta-molecules in the strong and weak bond regimes are presented on the left and right centers, respectively. $f_1$ and $f_2$ represent the two resonance frequencies of the meta-molecules in the strong bond regime. **d** Power spectra of the output waves at various $t_{pp}$. The blue-shaded region and the red curve represent the power spectra of the input and output waves, respectively.

We take into account the decay of the meta-atoms with the Lindblad operators. For an arbitrary operator of $Q$, we use[17,50]

$$\frac{\partial Q}{\partial t} = i[H,Q] + L\{\kappa_a + \kappa_J(t), a\}Q + L\{\kappa_b + \kappa_J(t), b\}, \quad (3)$$

where the Lindblad operator takes the form $L\{\kappa,O\}Q = \kappa(2O^\dagger QO - O^\dagger OQ - QO^\dagger O)$, and replaces the operators with their mean values: $\alpha = \langle a \rangle$ and $\beta = \langle b \rangle$. Then, we obtain the Langevin equations as follows,

$$\dot{\alpha} = -\left[if_a + \kappa_a + \kappa_J(t)\right]\alpha(t) - iJ(t)\beta(t) + \sqrt{2\kappa_e}E_p(t),$$
$$\dot{\beta} = -\left[if_b + \kappa_b + \kappa_J(t)\right]\beta(t) - iJ^*(t)\alpha(t) + \sqrt{2\kappa_e}E_p(t). \quad (4)$$

When the coupling is strong, the real part of $J$ is much larger than its imaginary part. We can study the dynamics of the eigen supermodes of the two meta-atoms, to a good approximation, on

the basis of $(a \pm ib)/\sqrt{2}$. For $f_a = f_b = f_0$, the eigen frequencies are $f_0 + |J(t)|$ and $f_0 - |J(t)|$. These supermodes can be driven with the same strength. To selectively excite one supermode, we tune the probe field to resonate with it.

In our device, the superconducting microbridges exhibit significant kinetic inductance in the superconducting state and transform into a lossy conductor in the normal state[51]. As a result, the bond strength becomes a time-varying complex variable. It can be described as a switching function that gradually changes over time:

$$sw(t, t_d, \tau) = [1 + e^{(t-t_d)/\tau}]^{-1}, \quad (5)$$

where $\tau$ is the duration determining the switching speed, and $t_d$ is the time when the amplitude drops to half in the switching process. Here, without loss of generality, we assume that the coupling coefficient, $J(t)$, between the two meta-atoms is modulated

in the time domain as

$$J(t) = i\,\mathrm{sw}(t, t_J, \tau_J) X_m + R_m [1 - \mathrm{sw}(t, t_J, \tau_J)] \qquad (6)$$

where $\tau_J$ is the duration for breaking the bond of the meta-molecules, $X_m = 0.167$ THz and $R_m = 0.03$ THz. Here, we set $\tau_J = 1.5\pi$ ps. The time evolution of $J$ in the calculation is shown in Fig. 1c. Before the bond breaks, we have $|J|_{max} = 0.167$ THz. After the bond breaks, the coupling is switched off, becoming $|J|_{min} = 0.03$ THz. The additional Ohmic loss can be assumed to be

$$\kappa_J(t) = \kappa_0 [1 - \mathrm{sw}(t, t_J, \tau_J)] \qquad (7)$$

with $\kappa_0 \approx 0.02$ THz. Before the bond breaks, the loss is zero. It becomes $\kappa_0$ after the bond is broken by the pump pulse.

In practical terms, the meta-molecule transitions from an initial strong bond state to a weak bond state under the THz pump. We calculated the transmission spectra in the strong and weak bond regimes, as depicted in the left and right center of Fig. 1c. In the calculation, the two resonators are assumed to be identical, with $f_a = f_b = f_0 = 0.47$ THz. In the strong bond regime, two resonant dips are located at 0.31 THz ($f_1$) and 0.63 THz ($f_2$) due to the resonance hybridization. The frequency interval of two dips is determined by the bond strength. When the center frequency of the input signal is 0.34 THz (approaching $f_1$), electrons can be resonantly excited from the ground state to the excited state when the probe pulse reaches the meta-molecules at time $t_p$. Upon the THz pump pulse reaching $t_J$, it breaks the bond of the meta-molecule. This breaking process takes approximately $\tau_J$. At this point, the two energy levels resulting from the resonance splitting become closer. When $t > t_J + \tau_J$, the bond completely breaks down, ideally resulting in two isolated meta-atoms. In our experiment, the two meta-atoms become weakly bonded, and there is only a single resonant dip at 0.50 THz (approaching $f_0$). Because the bond breaks rapidly, the energy-shifted excited states transition to the ground state, emitting photons at new frequencies around $f_0$. This process represents frequency conversion.

The equation for the narrowband THz probe pulse is:

$$E_p(t) = A\mathcal{E}(t) \cos[f_{in}(t - t_p)] \qquad (8)$$

where $A$ and $\mathcal{E}(t)$, are the amplitude and envelope of the pulse, respectively. $f_{in}$ is the center frequency of the input signal. According to our experimental observation, the carrier envelope can be assumed as follows:

$$\mathcal{E}(t) = [1 - \mathrm{sw}(t, t_p, \tau_{rise})] e^{-\left(\frac{t - t_p}{\sqrt{2}\tau_p}\right)^2} \qquad (9)$$

where $\tau_p$ and $\tau_{rise}$ together characterize the duration and varying rate of the input pulse (see Supplementary Fig. S4 for the calculated input pulse). In the model, we set $t_p$, $\tau_p$, and $\tau_{rise}$ as $50/2\pi$ ps, $27/2\pi$ ps, and $2/2\pi$ ps, respectively. The calculated power spectra of the input (shaded regions) and output pulses (red lines) are depicted in Fig. 1d. It is important to note that the shapes of the pump and probe pulses are not unique. The theoretical results may exhibit slight variations when different functions for $J(t)$ and $\varepsilon(t)$ are utilized. However, the underlying physics remains the same.

According to our model, three distinct cases are illustrated in Fig. 1c, d. When the tail of the probe pulse reaches the meta-molecules well ahead of the pump pulse ($t_{pp} < -14.0$ ps), the bond between the two meta-atoms remains time-invariant and strong for the duration of the probe pulse. It results in inefficient frequency conversion, and the output power spectrum (e.g., $t_{pp} = -16.0$ ps) lies entirely within the range of the input spectrum. When the pump pulse reaches the

meta-molecules before the probe pulse ($t_{pp} > 0$), the bond is broken (although weak in practice) prior to the meta-molecule being excited by the probe field. Subsequently, the probe, arriving later, only perceives two isolated meta-atoms. The excitation of the meta-atoms is minimal due to the process occurring predominantly off-resonance. In this scenario, the NbN microbridge is in a quasi-steady state after the pump pulse, and no frequency conversion takes place. When $t_{pp} = 0$ ps, the output spectrum lacks any new frequency component. Highly efficient frequency conversion occurs when the probe field reaches the metasurface ahead of the pump pulse but with a slight time difference. Initially, the probe field resonantly excites the ground state of the meta-molecule to the corresponding excited state. Subsequently, the intense THz pump rapidly breaks the meta-molecule into two weakly bonded meta-atoms. In this case, the excited states become degenerate in energy. Ultimately, the frequency conversion is achieved as the two meta-atoms emit photons at the new frequency $f_0$. For instance, the output spectra clearly reveal a newly converted component when $t_{pp} = -5.0$ ps.

## Phase controllable frequency conversion in time-varying meta-molecules

Figure 2a presents the microscopic image of the fabricated device. To investigate the time-varying response of the hybrid meta-molecule array, we used the THz pump-THz probe spectroscopy for measurement (see Supplementary Fig. S2 for the experimental setup). A multi-cycle THz probe pulse with a center frequency of 0.34 THz was obtained by letting the THz input pulse pass through a bandpass filter. The electric field direction of the probe pulse is perpendicular to the gap of the split ring resonator. The probe pulse passing through a bare MgO substrate was measured as a reference, and its power spectrum is depicted in Fig. 2b.

The mapping of the measured output spectra as a function of $t_{pp}$ is plotted in Fig. 2c. When $-8.0$ ps $\leq t_{pp} \leq -1.7$ ps, a new peak emerges on the right side of the main peak. Its amplitude is significantly higher than that in the reference spectrum. Considering the center frequency and bandwidth of the incident pulse in the experiment, we performed calculations and plotted the corresponding calculated spectra in Fig. 2d. The red marker curves indicate the converted frequency region. The blue arrows indicate the center frequency of the input pulse. The black dashed lines indicate the moments when the frequency conversion occurs and ends. The calculated results show good agreement with the measured spectra, validating the frequency conversion.

The spectral responses of the converted waves at different $t_{pp}$ are shown in Fig. 2e. When $t_{pp} = -10.0$ ps. There is only one primary peak at 0.37 THz in the power spectra. As $t_{pp}$ increases from $-8.0$ ps, new frequency components appear around the resonant frequency in the weak bond regime, specifically the resonance valley after the transition process. In addition to changes in amplitude, the peak frequency of the converted wave shifts, as shown by the red dashed line in Fig. 2e. We calculated the efficiency of the conversion peak as the ratio of the peak power of the converted wave to the input wave. The maximum efficiency of the conversion peak is 15%. We also evaluated the conversion efficiency by integrating the power of frequency conversion components and the total input power. The maximum conversion efficiency obtained by the experimental results is 4.1% (see Supplementary Note 7 for more details). When $t_{pp} = -1.5$ ps, there is only one main peak at 0.33 THz. The calculated output power spectra with the same $t_{pp}$ in Fig. 2e are presented in Fig. 2f, demonstrating good agreement with the experimental results.

We further analyzed the relationship between the input and the converted wave power. When $t_{pp} = -3.7$ ps, we obtained the output power spectra for various input powers ($P_{in}$), as shown in Fig. 3a. We analyzed the converted peak power at 0.47 THz (indicated as "A-E" in Fig. 3a) as a function of the input power. Figure 3b demonstrates that

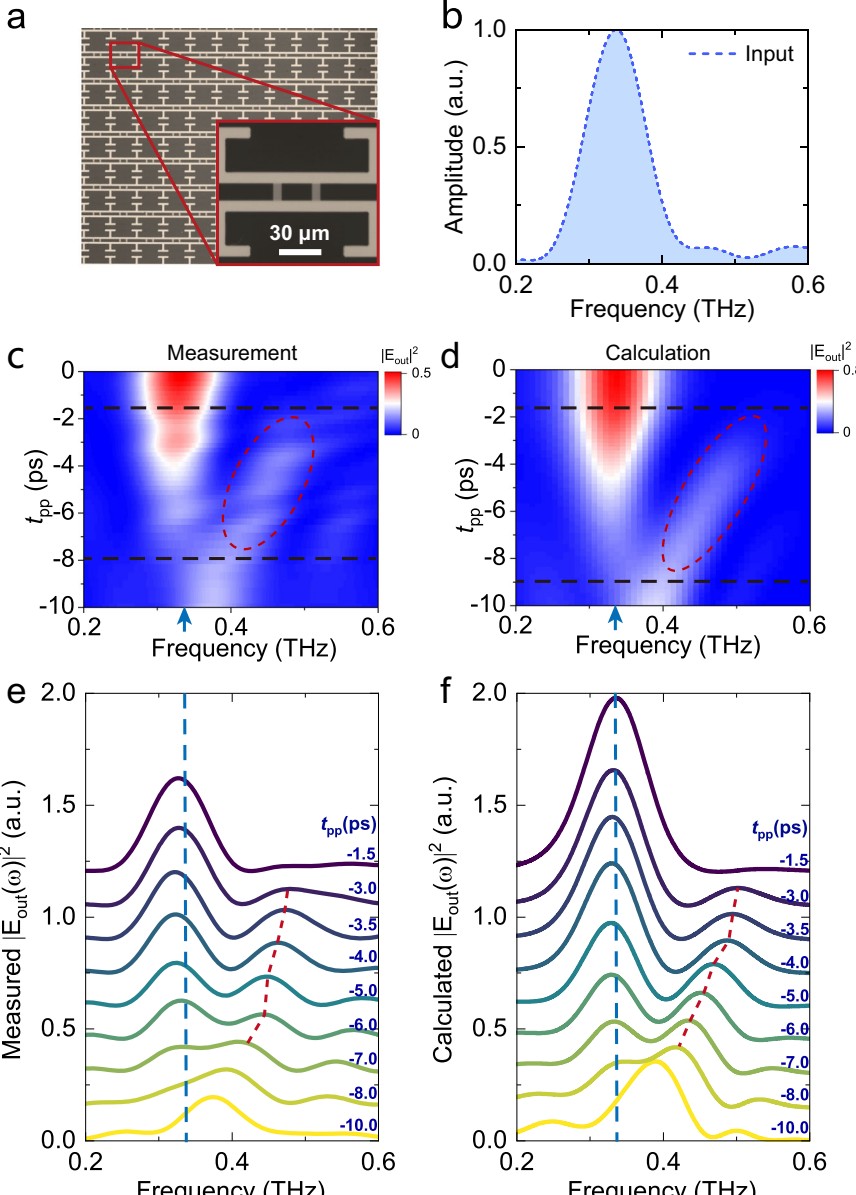

**Fig. 2 | Experimental and calculated spectra of frequency conversion in the hybrid meta-molecule array. a** Microscopic image of the fabricated superconductor-metal hybrid meta-molecule array. **b** Normalized reference power spectra obtained after the input pulse, centered at 0.34 THz, transmitting through a magnesium oxide (MgO) substrate. Two-dimensional plot showing the measured **c** and calculated **d** output power spectra as a function of the pump-probe delay ($t_{pp}$). The red marker curves indicate the converted frequency region. The blue arrows indicate the center frequency of the input pulse. The black dashed lines indicate the moments when the frequency conversion occurs and ends. **e** Measured and **f** calculated output power spectra for various $t_{pp}$ (each curve is vertically offset by 0.15). The red dashed lines indicate the frequency shift of the converted wave peak. The blue dashed lines indicate the center frequency of the input signal.

the input and converted wave powers follow a linear relationship, providing evidence of linear frequency conversion. We extracted the peak frequencies of converted waves ($f_c$) at different $t_{pp}$ from the measured and calculated spectra. In Fig. 3c, we observe a blueshift of $f_c$ in both the experimental and calculated spectra. The measured $f_c$ ranges from 0.42 THz to 0.5 THz as $t_{pp}$ increases from −8.0 to −1.7 ps, resulting in a frequency tuning range of approximately 80 GHz. In contrast, the calculated $f_c$ gradually increases from 0.42 THz to 0.52 THz, with a tuning range of about 100 GHz. Compared to previous reports[39], our frequency tuning range is larger due to the variation in the time derivative of $J(t)$ with $t_{pp}$, as illustrated in Eq. 6 and Fig. 1c. Under the THz pump, the complex conductivity of the superconducting microbridge undergoes significant changes, leading to $t_{pp}$-dependent variations in the coupling coefficient. The frequency

conversion process relies on the time derivative of the transition function around the time boundary. Due to the variations in the frequency components in the Fourier series expansion, the shift of the conversion frequency occurs.

We also conducted an analysis of the relative phase ($\Delta\varphi$) of the converted wave at the peak frequencies (corresponding to Fig. 3c) with respect to the input wave at the center frequency. The measured and calculated $\Delta\varphi$ as a function of $t_{pp}$ are shown in Fig. 3d. The measured $\Delta\varphi$ roughly exhibits a linear increase from nearly 0 to 1.7π as $t_{pp}$ increases from −8 ps to −1.7 ps. Although accurately measuring the phase coherence in the frequency conversion process is experimentally challenging, the converted field still clearly retains phase information of the input field. The prediction of the theoretical model is in good agreement with the experimental results, also strongly

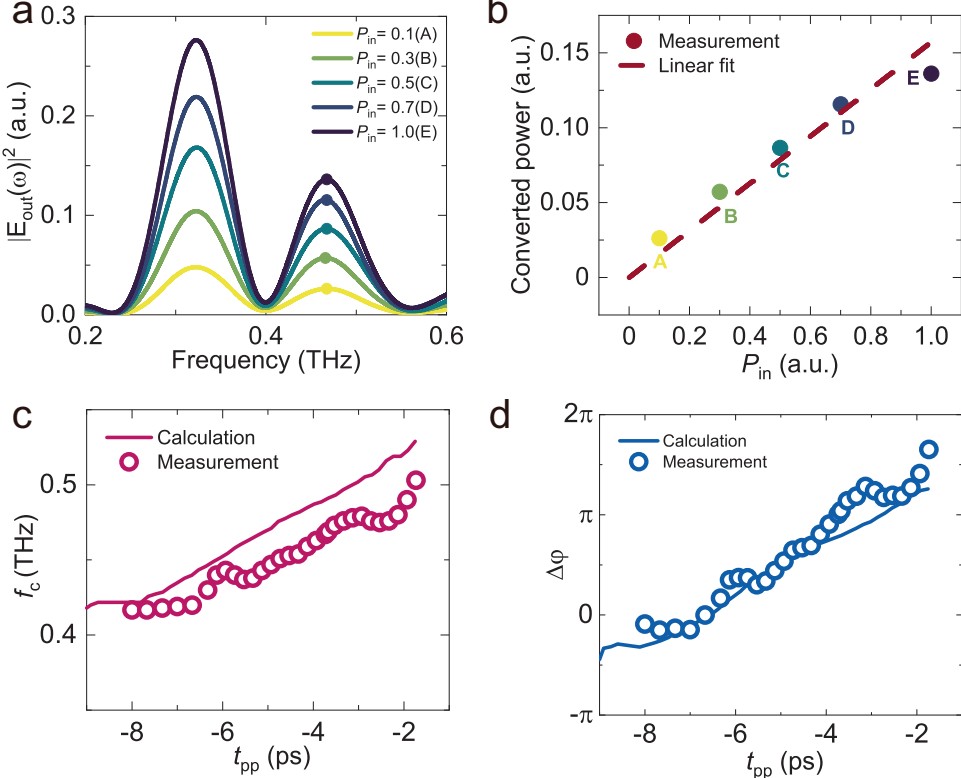

**Fig. 3 | Frequency and phase of the converted THz waves. a** Power spectra obtained under different incident powers ($P_{in}$) when $t_{pp} = -3.7$ ps. **b** Linear relationship between the converted power and the input power at 0.47 THz. The measured data are depicted with solid circles corresponding to the points in (**a**). The dashed line represents the fitted curve. **c** Frequencies of the converted wave peaks ($f_c$) as a function of $t_{pp}$. The measured $f_c$ is denoted by open circles, while the solid curve represents the calculated $f_c$. **d** The relative phase ($\Delta\varphi$) of the converted wave at the peak frequencies (corresponding to Fig. 3c) with respect to the phase of the input wave at the center frequency. The open circles show the measured $\Delta\varphi$, and the solid curve shows the calculated $\Delta\varphi$.

confirming the phase controllability of the frequency conversion. In this sense, the phase control manifests coherence in the frequency conversion process.

Additionally, we measured the output power spectra when the center frequency of the input wave was set to 0.6 THz (approaching $f_2$). We also observed linear frequency conversion in the time-varying region (see Supplementary Figs. S8, S9), with a new frequency component around 0.48 THz. The transmission spectra were in good agreement with the simulation results. However, we did not observe the linear frequency conversion phenomenon when the input wave was centered at 0.5 THz. These results indicate that the spectral response of the meta-molecules and the central frequency of the input wave influence the conversion efficiency.

We experimentally observed the tuning of the conversion efficiency with the applied voltage, which can be attributed to the electric heating effect (see Supplementary Note 10 for details). The electric tuning offers a more accessible method to control the output of the converted waves, which has significance for the application of the time-varying metasurface. In recent work, the highly efficient linear frequency conversion at the temporal boundary was experimentally demonstrated, which was attributed to the high Q factor of the resonant cavity[42]. We also found that the loss of metasurface is a crucial factor affecting the conversion efficiency (see Supplementary Notes 10–12 for details).

## Discussion

The efficient achievement of linear frequency conversion is attributed to the time-varying bond strength between meta-atoms induced by the THz pump. Both theoretical and experimental results indicate that the spectral response of the meta-molecules when the probe

pulse arrives at and after the transformation strongly influences the conversion frequency, phase, and amplitude. Consequently, adjusting the structure of the meta-molecules can effectively control the frequency conversion effect, and geometry optimization may further enhance the conversion efficiency. The introduction of electrical tunability further facilitates the control of linear frequency conversion.

The availability of ultrafast tunable metasurfaces offers a versatile platform for investigating various aspects of physics in the THz regime, including topological phenomena[52–54] and non-Hermitian physics[55,56]. Through experimental manipulation, different bonding configurations and tunable bonds between the meta-atoms can be realized. The developed time-varying coupled mode model serves as a valuable tool for comprehending the underlying physics. By employing the demonstrated experimental technique and the theoretical model, a deep exploration of the temporal transition of one- and two-dimensional topological states and phases becomes possible. If we periodically modulate the coupling in the topological metamaterials, we can study topological Floquet physics. We envisioned the extended research work in topological dynamics and Floquet physics based on the proposed metasurface (see Supplementary Note 13 for details). Additionally, by dynamically tuning the bonds, it is feasible to study the dynamics of non-Hermitian THz meta-molecules, such as the topological transfer of excitation around exceptional points. The linear frequency conversion via a temporal boundary can also be extended to other classical systems such as waveguides[41]. We envisioned the application enabling efficient on-chip frequency conversion (see Supplementary Note 14 for details).

The mechanism of THz radiation from our time-varying metasurface is essentially different from the quantum vacuum radiation

originated from the dynamical Casimir effect, which has been well interpreted with a quantum theory[57,58]. In the dynamical Casimir effect, the quantum ground state of a light-matter or two-mode coupled system is initially excited by the ultrastrong coupling via the anti-resonant interaction. When the boundary of the system is suddenly changed, the ground state changes and then radiates photons included in its excitation. This radiation is purely a quantum process created from the excited quantum vacuum state. In contrast, the converted field is generated via a classical process in our experiment, although our system is also time-varying. One supermode of the meta-molecule is initially excited by the classical THz probe field. Then, the coupling of the meta-atoms is rapidly tuned off. In this case, the energy of the excited supermode shifts, and the excited supermode radiates at a new frequency. The radiation from the quantum vacuum state is negligible here.

Our study demonstrates highly efficient linear frequency conversion in the THz regime utilizing superconductor-metal hybrid meta-molecules. Through extensive measurements of the output spectra at various pump-probe delays, we have validated our findings with the time-varying coupled mode model, establishing the significance of ultrafast breaking of the bond between meta-atoms as the underlying physical mechanism. Moreover, our investigation reveals the potential of electrical tuning in the metasurface, offering a convenient means to adjust the converted signal. The utilization of hybrid meta-molecules with time-varying boundaries presents a novel pathway for dynamic beam steering, pulse shaping, and the development of tunable THz sources.

## Methods

### Device fabrication
The NbN film, with a thickness of 12 nm, was deposited onto a 500-μm-thick MgO substrate using radio frequency magnetron sputtering. The superconducting transition temperature of the NbN film was measured to be 13.5 K. Subsequently, the NbN microbridges were patterned using ultraviolet photolithography and reactive ion etching with a mixture of $SF_6$ and $CHF_3$. The complementary pattern of the metallic resonators was formed using ultraviolet photolithography. A 200-nm-thick gold film was then deposited using radio frequency magnetron sputtering. Following the lift-off process, the sample fabrication was completed.

### THz spectroscopy measurement
A cryogenic THz pump-probe spectroscope system was utilized to measure the transmitted THz signal under an intense THz pump (see Supplementary Note 3 for details). The delay $t_{pp}$ could be adjusted by modifying the optical path difference between the THz probe and THz pump pulses. By changing the gate delay time ($t_{gate}$), the time-domain profiles of the transmitted THz probe pulse were recorded. After performing Fourier transforms, the THz transmission spectra through the sample were obtained.

## Data availability
The authors declare that all relevant data are available in the paper and Supplementary Information, or from the corresponding author on request.

## Code availability
The custom codes support the current study are available from the corresponding authors on request.

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

## Acknowledgements

This study was supported by the National Key Research and Development Program of China (Grants No. 2021YFB2800701, 2019YFA0308700); the National Nature Science Foundation of China (Grants No. 62288101, 62071217, 62027807, 62035014, 11890704, and 92365107); the Innovation Program for Quantum Science and Technology (Grant No. 2021ZD0301400); the Program for Innovative Talents and Entrepreneurs in Jiangsu (Grant No. JSSCTD202138); the Fundamental Research Funds for the Central Universities, and Research Fund for Jiangsu Key Laboratory of Advanced Techniques for Manipulating Electromagnetic Waves. J. W. acknowledges the support from Xiaomi Foundation.

## Author contributions

J.W., B.J., and G.Z. conceived the idea. S.D and J.W. designed the device. S.D. and X.J. fabricated the device. S.D., H.Q., and C.Z. built the experimental setup. S.D. and Y.J. performed the measurement. K.X., S.D., and X.S. performed the theoretical analysis. K.F., C.Z., G.Z., L.K., H.W., X.W., J.C., and P.W. provided a constructive discussion on the idea and the manuscript. S.D., J.W., and K.X. wrote the manuscript with input from all the authors. B.J. supervised the project.

## Competing interests

The authors declare no competing interests.
