## [Peer Review File NEW · Nature Communications]

Linear and phase controllable terahertz frequency conversion via ultrafast breaking down the bond of a meta-molecule.Reviewer #1 (Remarks to the Author):

Duan and coworkers present a combination of experiments and numerical simulations to demonstrate THz frequency conversion in a time-varying metasurface. The concept itself isn't new and the authors have cited prior works in this area (37,38). However, the platform used to introduce time-dependence to the metasurface and convert the THz frequency is new. The authors connect two metallic resonators with superconducting strips. The coupled resonators exhibit two resonant frequencies. The authors break this coupling by irradiating the superconductors with an intense THz beam. The resonators are then de-coupled and have only a single resonant frequency. If system is initially excited into the lower energy resonance of the coupled resonator and the bond is broken at the right time, the system can undergo free-induction decay at the shifted frequency of the uncoupled system. The measurements themselves are a standard application of THz time-domain spectroscopy. Therefore, I don't have any concerns about the validity of the results. Below are the main issues that I would like the authors to address:

-Although the work is a new variation of time-dependent metasurfaces, I'm concerned about how useful it actually is for THz frequency conversion. The THz pump pulse has a peak field of 25 kV/cm, which requires a relatively intricate optical system to produce. Is this really necessary to "break the bond"? If not, what is the threshold value and is this more accessible to those who may not have access to amplified 800 nm pulses?

-On a related topic, the frequency shift is relatively small and is determined by the design of the metasurface. The new frequency is also present in the THz pump pulse. What would happen if you simply applied a linear filter to the pump pulse at the target frequency (0.48 THz)? This would also be a linear process and I suspect that it would be more efficient than the approach used by the authors. This also leads me to think that this approach is not very practical.

-Ultimately, it would be desirable to shift to frequencies that were not already present in the original THz pulse. Can the authors comment on whether this is feasible?

-For Figure S2: where is the optical chopper or other modulation device located? The authors use polarization filtering to ensure that only the converted THz frequency from the "probe" and not any excitation from the "pump" is detected. This seems to work, which I find a little bit surprising, given that the resonators should radiate a finite cross-polarized component. However, the position of the chopper could be used to further isolate the detected THz pulse. Was this also used?

-The authors currently define the conversion efficiency in terms of the power of the input, filtered THz beam and the output, frequency-shifted THz beam. I think that it's important to also quote the overall optical conversion efficiency. What is the total power of THz incident on the metasurface and what is the power of the frequency-shifted THz? Please calculate a conversion efficiency parameter based on this as well.

-The authors emphasize that, because they use an intense THz pulse to break the bond, it is a coherent process. However, I don't feel that it is any more coherent than if they had used an 800 nm pulse to break the bond. The main coherent feature is free inductive decay and this will be there regardless of how the bond is broken. Can the authors comment on what coherent features are unique to their measurement approach and would not be possible if an 800 nm pulse was used instead? It would be nice if the authors could repeat this measurement using an 800 nm pulse to break the bond.

-What does the waveform/spectrum of the transmitted pump pulse look like? I would be interested to see this as well and whether there is any evidence of the bond breaking on this pulse.

A few other minor details:

-Please remove the word "Novel" from the abstract.

-Please include error margins on all of your plots.

-The choice of colormap for Figs 2 (c) and (d) limits how well the figures can be analysed by eye. In particular, they look very saturated. Please use a different colormap. As well, the colormap label, "Amplitude" does not clearly indicate whether this is for the electric field or power of the waveform.

In summary, I feel that the work is well done. However, since THz frequency conversion in a time-varying metasurface has already been demonstrated, the novelty is somewhat limited. As I expressed above, I also have concerns about whether the technique will actually be useful. I don't recommend publication at this point, but would be happy to reconsider if the authors can address my comments above.

Reviewer #2 (Remarks to the Author):

In the submitted manuscript, the authors propose an intriguing exploration of coherent linear THz frequency conversion using a time-varying metasurface platform. The unit cell, termed as a metamolecule, is designed from two split-ring resonators connected by superconducting bridges.

Upon excitation by an intense THz pulse, the bonds inside the metamolecule (via the bridges) become weakened quite rapidly, causing the metamolecule to split into two metaatoms. This abrupt temporal change, also referred to as the time interface, results in the spectral conversion of the incoming THz pulse.

The results presented in the manuscript, while showing potential, have some parallels with previously published work [Ref. 37]. The manuscript differentiates itself through the introduction of time-varying elements (in this case, superconductors), the choice of THz frequency for time-varying excitation of the platform, and the implementation of electrical control on the spectral conversion.

However, it would be beneficial if the authors could elaborate further on the advantages of their approach. Offering additional context, implications, and potential impacts of their platform could improve the presentation of the manuscript.

Reviewer #3 (Remarks to the Author):

The authors propose a method of frequency conversion in a time modulated meta-surface, where two split-ring resonators are connected with superconducting bridges. A pump pulse allows controlling the superconducting element, which can be either inductive or purely resistive, depending on the temperature rise induced by the pulse. As a result, the electromagnetic resonance of the system is altered switching between two configurations: coupled resonator case, with two different frequencies ω_1 and ω_2 and uncoupled resonator case with a single frequency ω_0 . The authors perform pump-probe experiments where they study the dynamics of the switching regime, and interpret their result in terms of coherent frequency conversion.

This work clearly bears strong inspiration from a prior work by Lee and al. that was published in Nature Photonics, <https://doi.org/10.1038/s41566-018-0259-4> (ref. 37): a fact acknowledged by the authors. In particular, their explication of the frequency conversion effect resembles strongly to the concept of "temporal boundary" introduced in Ref. 37. Nevertheless, the meta-surface device proposed by the authors operates on a quite different principle. Also, the experimental data and the corresponding modelling seem to be much more clear than the ones in Ref. 37. Indeed, in the present case there is a clear pic that corresponds to the frequency converted signal, whereas in the data of Ref. 37 the spectral features were fairly broad (Fig. 3a in Ref. 37). The paper thus seems to contain sufficient novelty and could, in principle, be published in Nature Communications, however it still lacks sufficient details, clarity, and even scientific insight. Some (including major) corrections are needed:

a. It will be very helpful for the reader if the authors indicate clearly the "original frequency" and the "converted frequency" , f_c , in Figures such as 2(c,d,e, f). Is the original frequency the one 0.34 THz?

b. How was the converted power in Fig. 3b estimated: is it the peak power at the frequency f_c (dots in Fig. 3a), or some integration was involved?

c. In the introduction, the authors claim a possibility of applications, however this claim seems overstatement as their device is operated at 4K.

d. The authors talk about "coherent frequency conversion", and they present data and simulation on the phase difference in Figure 3d. In the simulation, there is a strong 2π -jump of the phase, indicating much stronger variations than in the experiment. Also, the experimental phase at -8ps is almost zero, while the simulation predicts a value of almost $\pi/4$, which is a significant difference. The trends for the simulated and measured phase is fairly different. The claim that the authors observe a coherent effects is thus strongly overrated, as the experimental phase behaves very differently from the model; that claim should be removed from the title and the introduction.

e. The electrical measurements in Figure 4 present an interesting add-on but, besides the description of the device and a few measurements, this configuration is not very well exploited and not sufficiently commented. This is clearly a means for a static control of the meta-surface, the physical effects being rather trivial (killing the superconductivity by Joule heating), but what is the point of that? What does this static mechanism has to do with the time-varying modulation and what do we learn from this? This part can be removed; unless an experimental demonstration of a new effects (i.e. applying an RF bias to periodically modulate the coupling constant by all electrical means).

f. The model contained in Eq. (1) is pretty interesting, and seems to explain pretty well the data, but it has not be detailed enough, not even in the supplementary materials. It will be very interesting if the authors could provide more details and developments based on the Hamiltonian (1), and probably provide analytical expressions for the frequency and phase shift in certain simplified cases. Also the physics contained in the Hamiltonian is similar to the one used to describe the dynamical Casimir effect (see <https://journals.aps.org/prl/abstract/10.1103/PhysRevLett.98.103602>) , the authors could comment their work within this context.

g. It will be also helpful if the authors could comment on the selection rules for their structures. For instance, because of the symmetry of their meta-atom, the uncoupled mode should be degenerate into two modes, a dark mode, where the currents in the inductive loops oscillate in phase opposition, and a bright mode, where they oscillate in phase. Also, similar rules should apply for the coupled system, where one of the coupled modes should be much brighter than the other. This means that not only the coupling strength J is modulated, but also the radiation loss (parameters k_a and k_b in Eq.(1)). Have the authors taken into account this effect, that can be quite significant, and how does it affect the data modelling?

h. The authors should definitely provide field maps illustrating the resonant modes.

-The overreach of this work should be expanded, for instance what type of effects related in the topological dynamics and Floquet physics can be expected? How shall the structure designed by the authors be modified in order to make such effects observable?

Dear Editor and Reviewers,

We would like to thank you for your efforts on our manuscripts. We greatly appreciate the constructive and valuable comments from the reviewers. We have carefully addressed the comments and queries raised by the referees (comments from the referees are in blue italicized). Point-by-point responses to the reviewers are listed below in this letter. Besides that, we attach an additional manuscript in which the major text changes are marked in red. We believe that the revised manuscript has taken full account of the points raised by the reviewers, and we hope you will be satisfied with the revised version of this paper.

Best regards,

The authors

Reviewer #1

Duan and coworkers present a combination of experiments and numerical simulations to demonstrate THz frequency conversion in a time-varying metasurface. The concept itself isn't new and the authors have cited prior works in this area (37,38). However, the platform used to introduce time-dependence to the metasurface and convert the THz frequency is new. The authors connect two metallic resonators with superconducting strips. The coupled resonators exhibit two resonant frequencies. The authors break this coupling by irradiating the superconductors with an intense THz beam. The resonators are then decoupled and have only a single resonant frequency. If system is initially excited into the lower energy resonance of the coupled resonator and the bond is broken at the right time, the system can undergo free-induction decay at the shifted frequency of the uncoupled system. The measurements themselves are a standard application of THz time-domain spectroscopy. Therefore, I don't have any concerns about the validity of the results. Below are the main issues that I would like the authors to address:

Reply: We greatly appreciate the reviewer for the positive comments and the comprehensive summary of our work. Although prior works have reported THz frequency conversion using a time-varying metasurface, a major contribution of this work is providing a “new platform” for studying time-varying metasurface, as stated by the reviewer. We are also very grateful to the reviewer for acknowledging the validity of our results. It recognizes the efforts we have put into building the THz pump-THz probe time-domain spectroscopy system and many measurements. We have addressed all the comments raised by the reviewer in the replies below. We hope the reviewer can recognize us for our efforts to improve the work.

-1_1. Although the work is a new variation of time-dependent metasurfaces, I'm concerned about how useful it actually is for THz frequency conversion.

Reply: We thank the reviewer for pointing out our work reports a new time-dependent

metasurface. The comment on “use” also encourages us to think about the significance and application of this study.

Recently, the study on the time-varying electromagnetic system has attracted much attention. The introduction of time dependence has exhibited various promising applications such as topological phase transition (*Phys. Rev. Lett.* 125, 013902(2020), *Phys. Rev. Lett.* 125, 166801(2020)), nonreciprocity, and Floquet phenomena. In this work, we establish a complete theoretical and experimental platform for studying time-varying systems. It provides a good basis for subsequent applications. We believe that this platform will be applied in the following two directions.

One envisioned application is to achieve on-chip frequency conversion. The schematic of the on-chip frequency converter based on a single time-varying meta-molecule is shown in Fig. R1. We chose a single wire transmission line that lies on top of a supporting dielectric substrate, known as the planar Goubau line (PGL) (*Phys. Rev. X* 1, 021016 (2011)), for guiding the electromagnetic waves. The meta-molecule interacts with the PGL through the near-field coupling. The coplanar waveguide (CPW) guides the THz pump wave used to “break the bond” of a meta-molecule. Compared to the meta-molecular arrays, this approach reduces the energy of the THz pump wave by several orders. Thus, it can significantly reduce energy consumption and make the device more compatible.

Fig. R1. On-chip frequency converter consisting of a single time-varying meta-molecule.

Another application of the time-varying metasurface is in topological photonics. We have provided two specific designs of the topologic metasurface to study the novel phenomenon in the revised supplementary material (see Supplementary Note 13 for details). We used superconducting microbridges to couple two meta-atoms, enabling temporal tuning of their coupling strength (J) with intense THz pulses. This platform allows us to study temporal dynamics in 1D topological chains and 2D topological metasurfaces. In the 1D chain, meta-atoms A and B are coupled with J_1 within unit cells and J_2 between unit cells, forming a Su-Schrieffer-Heeger (SSH) model. We aim to explore the temporal dynamics of SSH topology in the THz regime. Additionally, this configuration has potential in 2D topological metasurfaces, with topology dependent on J_1 , J_2 , and J_3 (representing upper and lower meta-atom coupling). Dynamic tuning of three coupling rates allows the investigation of temporal transitions between different topological phases and the emergence of edge and corner states in these configurations.

-1_2. The THz pump pulse has a peak field of 25 kV/cm, which requires a relatively intricate optical system to produce. Is this really necessary to “break the bond”? If not, what is the threshold value and is this more accessible to those who may not have access to amplified 800 nm pulses?

Reply: Thank the reviewer for the valuable comment. We show the new frequency conversion mechanism is required to break the bond on the picosecond timescale, which is comparable to the oscillation period of the THz probe pulse. The breaking of the bond results in ultrafast and remarkable changes in the spectral response of the meta-molecule, which is critical to achieving highly efficient conversion. Accordingly, “break the bond” is necessary.

As for the experimental setup generating the intense THz pump, it has been a mature technique widely used in THz spectroscopy studies to explore the nonlinear effects of materials and devices. In our work, such a strong THz electric field is required to “break the bond” of the meta-molecule. In the previous work of our group, we demonstrated that the superconducting property of a 15 nm thick NbN film is

completely suppressed when the THz field strength reaches 18 kV/cm. The 12 nm-thick NbN film is used in our designed time-varying metasurface, and the field strength used to break the bond was 25 kV/cm, which is comparable to the previous studies.

I guess the reviewer is wondering whether this experiment can be done with an 800 nm femtosecond laser pump. It is not feasible for the femtosecond laser pump without an amplifier because the energy of a single pulse is low. Since the repetition frequency of the femtosecond laser without an amplifier is 80 MHz, the pulse energy is $\sim 0.01 \mu\text{J}$. Based on the previous experimental results, the pulse energy is too low to suppress the superconducting property of NbN film.

For our designed time-varying metasurface, the amplified 800 nm femtosecond laser pump can “break the bond” as well. However, the transition time of the superconducting microbridge under the optical pump is much longer than the intense THz pump based on recent references (Ref. 45-47). As reported in Ref. 47, the optical pump-THz probe (OPTP) measurement results show that the transition time from the superconducting state to the normal state is more than 4 ps, while the THz pump-THz probe (TPTP) measurement results show that the transition time is less than 1 ps.

The transition time of the microbridge from the superconducting state to the normal state determines the transition time of the coupling coefficient (J) in the time-varying coupled mode model. The transition time of J can be changed by tuning τ_J . The value of τ_J used in our theoretical model is denoted as τ_{J0} ($\tau_{J0} = 1.5\pi$ ps). We increase τ_J and calculate the frequency conversion effect. Figure R2a, b shows the time evolution of J and the calculated output power spectra when $t_{pp} = -6$ ps for different τ_J . The larger the τ_J , the longer the transition time of J is. It is evident that the longer the transition time of J , the lower the conversion efficiency is. Based on the above analysis, the THz pump pulse is more appropriate for achieving high-efficiency frequency conversion.

Fig. R2. (a) Time evolution of J for different τ_j . (b) Calculated input and output power spectra when $t_{pp} = -6$ ps for different τ_j .

Finally, we respond with the threshold energy needed to “break the bond.” In our experimental system, the THz power emitting from the LiNbO₃ crystal is 1 mW, which is measured by a THz power meter. At the focal point where the sample is located, the power is 0.5 mW. Due to the attenuation by the cryogenic Dewar, the THz power reaching the sample is about 0.056 mW. The energy of each THz pulse is 5.6×10^{-8} J. For the metasurface, there are 1965 meta-molecules covered by the spot of the THz beam. Our other measurement results show that using 1/8 of the maximum field strength can trigger the phase change of the microbridge. The energy to drive the phase transition of a single meta-molecule is estimated to be only 4.4×10^{-13} J.

-2. On a related topic, the frequency shift is relatively small and is determined by the design of the metasurface. The new frequency is also present in the THz pump pulse. What would happen if you simply applied a linear filter to the pump pulse at the target frequency (0.48 THz)? This would also be a linear process and I suspect that it would be more efficient than the approach used by the authors. This also leads me to think that this approach is not very practical.

Reply: We sincerely thank the reviewer for this valuable comment. We experimentally obtained a large frequency shift of 80 GHz with the change of pump-probe delay.

Based on our experimental results, the frequency shift is not only determined by the design of the metasurface but also by the pump-probe delay. The remarkable frequency shift of the proposed device can be attributed to the significant change in the complex conductivity of the superconducting microbridge under the THz pump.

Though the THz pump pulse in our experiment is broadband and contains the spectral component at 0.48 THz, it is not necessary for the pump pulse to contain the target frequency of 0.48 THz. The primary role of the THz pump pulse in the time-varying process is to “break the bond.” The energy of the converted wave comes from the input signal itself and is independent of the frequency components in the pump pulse. Moreover, the electric field direction of the pump pulse is perpendicular to that of the input pulse in the measurement. The detector crystal only detects the transmitted input pulse rather than the transmitted pump pulse.

As for the practicality of the approach, we admit that the current experimental setup based on LiNbO₃ crystal may be bulky and intricate for practical applications. We respect the reviewer's concerns about the practicality of frequency conversion, but this work aims to demonstrate a conceptually new mechanism for efficient frequency conversion in the THz regime. The potential applications of efficient frequency conversion have been showcased, which is significant for exploring exotic physics and device applications.

-3. Ultimately, it would be desirable to shift to frequencies that were not already present in the original THz pulse. Can the authors comment on whether this is feasible?

Reply: We sincerely thank the reviewer for this valuable comment. Shifting to the frequencies not already present in the original pulse is feasible. Ideally, we can use a single-frequency pulse as the input signal, and the conversion of the input frequency to the target frequency will occur at the time boundary.

In our experiments, due to the lack of a source capable of generating a THz single-frequency pulse, we let the broadband pulse pass through a narrowband filter to generate the narrowband THz pulse. However, the narrowband filters cannot

completely suppress the out-of-band signals. The truncation of the time window in the data processing also causes spectrum leakage for experimental and calculation results. Therefore, the input pulse inevitably has a component at the converted wave frequencies in experiments and simulations.

Although we cannot produce an input signal that does not contain components at converted wave frequencies, we can increase the frequency difference between the input and the converted waves in the theoretical calculation. In that case, the frequency components of the input signal at the converted wave frequencies significantly decrease. For example, we set the input signal to 0.27 THz in the calculation model and calculated the power spectrum of the output waves when $t_{pp} = -4$ ps, as shown in Fig. R3. As labeled by the purple circle, the power spectrum of the output waves still appears at the frequency where the corresponding amplitude of the input signal is almost 0.

Fig. R3. Calculated power spectra of the input pulse with a center frequency of 0.27 THz and of the output wave when $t_{pp} = -4.0$ ps.

-4. For Figure S2: where is the optical chopper or other modulation device located? The authors use polarization filtering to ensure that only the converted THz frequency from the “probe” and not any excitation from the “pump” is detected. This seems to work, which I find a little bit surprising, given that the resonators should radiate a finite cross-polarized component. However, the position of the chopper could be used to further isolate the detected THz pulse. Was this also used?

Reply: We sincerely thank the reviewer for this valuable comment. We are sorry that the optical chopper is missing in Fig. S2. The optical chopper has been added in the updated Fig. S2. As shown in Fig. R4 (copied from the updated Fig. S2), it is placed in the path of the femtosecond laser beam in front of the ZnTe crystal, which is used to generate the THz input pulse. The chopper has a modulation frequency of 370 Hz.

As mentioned by the reviewer, the optical chopper could be used to isolate the interference of the THz pump on the detection signal. The chopper modulates the input signal while the pump signal is unmodulated. In the signal acquisition process, we utilized a lock-in amplifier to extract the modulated signal, ensuring that the pump signal is excluded from the measurements.

Fig. R4. Diagram of the THz pump-THz probe spectroscopy system.

Thank the reviewer for reminding us that the cross-polarization transmission component of the pump wave may cause interference with the probe signal. In the measurement, the electric field directions of the pump and probe pulses are parallel and perpendicular to the gap of resonators, respectively. A polarizer is put in front of the ZnTe crystal to selectively detect signals with an electric field direction vertical to the gap. To clarify whether the resonator radiates a finite cross-polarized component, we

perform numerical simulations to calculate the co-polarization and cross-polarization transmission spectra of the metasurface.

Figure R5a shows the schematic of THz wave transmission through the metasurface. The direction perpendicular to the gap is defined as the y -direction. Figures R5b and 5c show the co-polarization (T_{yy}) and cross-polarization (T_{yx}) transmission spectra when the electric field direction of the incident wave is vertical to the gap. The maximum cross-polarization transmission coefficient is only 2×10^{-14} . In our experimental setup, the power of the input signal reaching the metasurface is roughly $1.0 \mu\text{W}$. Therefore, the cross-polarization transmission component of the probe signal is negligible.

Fig. R5. (a) Schematic of terahertz wave transmission through the metasurface. Simulated (b) co-polarization (T_{yy}) and (c) cross-polarization (T_{yx}) transmission spectra for the probe pulse. Simulated (d) co-polarization (T_{xx}) and (e) cross-polarization (T_{xy}) transmission spectra for the pump pulse.

We also perform the simulation when the electric field direction of the incident wave is parallel to the gap, consistent with the polarization direction of the pump pulse. Figure R5d,e displays the simulated co-polarization (T_{xx}) and cross-polarization (T_{xy}) transmission spectra. The power of the pump pulse at the metasurface is roughly 0.056 mW. The highest cross-polarization transmission coefficient for the pump pulse is 4.8×10^{-14} . Thus, the power of the transmitted cross-polarization transmission signal is 2.7×10^{-18} W. Based on the above calculation, the cross-polarization transmission component of the pump pulse is remarkably lower than the co-polarization transmission component of the probe pulse. Furthermore, introducing the chopper and the lock-in amplifier can further exclude the interference from the pump waves.

To fully address the valuable comment in the revised supplementary information, we have updated Fig. S2 and added the following sentences in the supplementary Note 2. *“The chopper is placed in the path of the laser beam to generate THz input signals. The chopper modulates the input signal with a modulation frequency of 370 Hz, while the THz pump wave is unmodulated. In the signal acquisition process, we utilized a lock-in amplifier to extract the modulated signal, ensuring that the inference from the THz pump is excluded from the measurements.”*

-5. The authors currently define the conversion efficiency in terms of the power of the input, filtered THz beam and the output, frequency-shifted THz beam. I think that it's important to also quote the overall optical conversion efficiency. What is the total power of THz incident on the metasurface and what is the power of the frequency-shifted THz? Please calculate a conversion efficiency parameter based on this as well.

Reply: We sincerely thank the reviewer for this valuable suggestion. The original conversion efficiency (η) was calculated as the ratio of the peak power of the converted wave (P_{conv}) to the input wave (P_{in}). Following the suggestion of the referee, we reevaluate the conversion efficiency by integrating the frequency conversion components of the output power spectra and the total power of the input wave.

The orange region in Fig. R6a is the frequency conversion region minus the area

in the corresponding frequency range of the input power spectrum. In this case, the conversion efficiency is defined as the ratio of output power beyond the input power spectrum to the total input power. The calculated conversion efficiency is 6.37% when $t_{pp} = -5$ ps. Figure R6b, c shows the conversion efficiency obtained from the calculated and measured transmission spectra. They exhibit a similar trend as t_{pp} increases. The maximum conversion efficiencies obtained from the calculated and measured transmission spectra are 6.4% and 4.1%, respectively.

Fig. R6. (a) Calculated input and output power spectra when $t_{pp} = -5.0$ ps. The orange region indicates the output power spectrum that is beyond the input power spectrum. Conversion efficiency as a function of t_{pp} calculated from the experimentally measured (b) and calculated (c) power spectra of input and output waves.

In the experimental setup, a laser power of 270 mW is incident on ZnTe crystal to generate the THz probe pulse. The conversion efficiency is around 0.1%, so the power of the THz probe pulse is approximately 27 μ W. After passing through a narrowband filter, the power is estimated to be around 9.4 μ W. Due to the attenuation of the cryostat window, the power incident on the metasurface is roughly 1.0 μ W. The maximum power of the converted THz waves is estimated to be 0.041 μ W.

In the revised supplementary information, we added the conversion efficiency results using the above method. Please refer to Supplementary Note 7 in the revised supplementary information.

We also amended the sentences in the third paragraph after Fig.2 in the revised

manuscript. “We calculated the efficiency of the conversion peak as the ratio of the peak power of the converted wave to the input wave. The maximum efficiency of the conversion peak is 15%. We also evaluated the conversion efficiency by integrating the power of frequency conversion components and the total input power (see Supplementary Note 7 for more details).”

-6. The authors emphasize that, because they use an intense THz pulse to break the bond, it is a coherent process. However, I don't feel that it is any more coherent than if they had used an 800 nm pulse to break the bond. The main coherent feature is free inductive decay and this will be there regardless of how the bond is broken. Can the authors comment on what coherent features are unique to their measurement approach and would not be possible if an 800 nm pulse was used instead? It would be nice if the authors could repeat this measurement using an 800 nm pulse to break the bond.

Reply: We sincerely thank the reviewer for drawing our attention to whether the frequency conversion process in our work is coherent. Based on our understanding, we think the frequency conversion process is a coherent process because there is a stable relative phase between the converted and input waves. Our experimental results and theoretical calculations confirm that the frequency conversion process is coherent.

We have updated Fig. 3d and refined the discussion on phase coherence in the revised manuscript. In the original Fig. 3d, we fixed the frequency of the converted wave at 0.47 THz. However, the peak frequency of the converted wave varies with t_{pp} . In the revised Fig. 3d, we plot the measured and calculated relative phase of the converted wave at the peak frequencies with respect to the phase of the incident wave at the center frequency. After correcting the method, we found that the theoretical and experimental results conformed better. We believe the refined results better illustrate the coherence of the frequency conversion process. Please refer to Fig. 3d in the revised manuscript.

For the 800 nm femtosecond laser pump, we agree with the reviewer that the physics of frequency conversion is similar. As stated in the previous reply, the 800 nm

femtosecond laser pump can also “break the bond,” and it is possible to observe the frequency conversion effect. However, the transition time of the superconducting microbridge under the optical pump is much longer than the intense THz pump. Based on our theoretical analysis, the frequency conversion efficiency will be much lower, which is verified in Fig. R2 of the previous reply. Moreover, the longer transition time may suppress the coherence between the converted and input waves.

-7. What does the waveform/spectrum of the transmitted pump pulse look like? I would be interested to see this as well and whether there is any evidence of the bond breaking on this pulse.

Reply: We sincerely thank the reviewer for this valuable comment. The time-domain waveform and power spectrum of the THz pump pulses are plotted in Figs. R7a and R7c, respectively. The duration of the THz pump pulse is approximately 3 ps, and it has a bandwidth from 0.2 to 1.2 THz. The maximum electric field strength of the THz pulse (E_0) reaches 25 kV/cm.

When only the pump pulse with the electric field perpendicular to the gap passes through the sample, the waveform of the transmitted pump pulse is plotted in Fig. R7b. The corresponding power spectrum of the transmitted pump pulse is shown in Fig. R7c. The THz transmission spectrum, as shown in Fig. R7d, is consistent with the calculated spectrum in the weak-coupling regime of Fig. 1c. It means the pump pulse can effectively cause the phase transition of superconducting microbridge from the superconducting state to the normal state, *i.e.*, “break the bond”. Based on the above analysis, the transmitted pump pulse spectrum provides evidence of “bond breaking”.

To fully address the valuable comment, we have updated Fig. S3 and added the above discussion in Supplementary Note 3. Please refer to the revised supplementary information.

Fig. R7. Measured time-domain waveform of the pump pulse (a) and the transmitted pump pulse (b). (c) Corresponding power spectra of the input and transmitted pump pulses. (d) Transmission spectrum which is the ratio of the power spectrum of the transmitted pump pulse to the input pump pulse.

A few other minor details:

-8. Please remove the word “Novel” from the abstract.

Reply: We sincerely thank the reviewer for this valuable suggestion. The word “novel” has been removed from the abstract in the revised manuscript. Please refer to the revised manuscript.

-9. Please include error margins on all of your plots.

Reply: We sincerely thank the reviewer for this valuable suggestion. Due to the peculiarities of the THz time-domain spectroscopy measurement, it is difficult to add

error margins to the data. The transmission spectra in our experimental work are obtained by the Fourier transform of the time-domain waveforms. Calculating the amplitude and phase errors in the transmission spectra is difficult from measuring the time-domain signals. We also checked the recent literature on THz time-domain spectroscopy measurement, and it is customary not to mark the error margins. Nevertheless, we ensure the reliability of the measurement results through the following operations in the experiments.

For THz time-domain spectroscopy experiments, measurement errors primarily stem from inherent instrument and environmental noise. In the actual experiments, we first optimized the optical setup to improve signal quality and boost the signal-to-noise ratio of the experimental system. After debugging, it is ensured that the repeated measurements of the time domain signals were stable and the waveforms were smooth without burrs and fluctuations.

In addition, regarding the detection principle of a THz time-domain spectroscopy system, the measured time-domain waveform is the average of multiple measurements. In our experimental setup, the pulsed laser has a repetition frequency of 1 kHz, and the integration time of the lock-in amplifier is set to 300 ms. It means that each value at each time point of the time-domain waveform is averaged over 300 pulses. The long integration time of the lock-in amplifier ensures a high signal-to-noise ratio. Therefore, the results of a single measurement using a calibrated THz time-domain spectroscopy system are also credible.

-10. The choice of colormap for Figs 2 (c) and (d) limits how well the figures can be analysed by eye. In particular, they look very saturated. Please use a different colormap. As well, the colormap label, "Amplitude" does not clearly indicate whether this is for the electric field or power of the waveform.

Reply: We sincerely thank the reviewer for this valuable comment. We regret choosing colors that were too saturated. We have updated Fig. 2c, d in the revised manuscript. It is copied in Fig. R8 below. A new set of gradient colors is chosen, and the converted

frequency region is circled with a red dashed curve. In the previous figure, the levels in the converted frequency region had little difference relative to the minimum level in the contour map, making it difficult to highlight this feature. We have adjusted the color scale to clarify this feature in the revised figure. Hopefully, the above adjustment can improve the visualization and clarity of the data.

Additionally, we have accepted the reviewer’s suggestion and changed the colormap label to “ $|E_{\text{out}}|^2$ ” to represent the power spectra.

Fig. R8 (updated Fig.2c, d). Two-dimensional plots of the measured (c) and calculated (d) output power spectra as a function of the pump-probe delay (t_{pp}).

In summary, I feel that the work is well done. However, since THz frequency conversion in a time-varying metasurface has already been demonstrated, the novelty is somewhat limited. As I expressed above, I also have concerns about whether the technique will actually be useful. I don't recommend publication at this point, but would be happy to reconsider if the authors can address my comments above.

Reply: We sincerely thank the reviewer for this positive comment. We admit the previous work about time-varying metasurface has done well, but there are still many aspects that need to be improved. Our work presents a new platform for THz frequency conversion with a completely different operating principle. The model explains the physical mechanism of linear frequency conversion more clearly. We believe we have made substantial progress in improving the conversion efficiency and phase

controllability. Our methodology can advance the study of the various exciting phenomena in time-varying medium and the application of the THz frequency converters.

Thank the reviewer for reminding us of the practicality of the proposed device. In our revised manuscript and supplementary information, we have added some discussion of device applications. As the reply to the first comment mentions, we give two application examples of on-chip frequency converter and topological photonics. The first envisioned application is realizing the on-chip frequency conversion based on a single time-varying meta-molecule. The efficient on-chip frequency conversion is essential for manipulating THz signals on a chip. Please refer to Supplementary Note 14 for more details. Furthermore, the proposed design can significantly facilitate integration and scalability. Another envisioned application is to study topological dynamics. The proposed design in the revised Supplementary Note 13 enables the study of temporal dynamics in 1D topological chains and 2D topological metasurfaces. The 1D chain forms a Su-Schrieffer-Heeger (SSH) model, which provides a nice platform to study the temporal dynamics of SSH topology in the THz regime. The proposed 2D topological metasurfaces with varying coupling rates offer a tool to explore novel phenomena.

Reviewer #2

In the submitted manuscript, the authors propose an intriguing exploration of coherent linear THz frequency conversion using a time-varying metasurface platform. The unit cell, termed as a metamolecule, is designed from two split-ring resonators connected by superconducting bridges.

Upon excitation by an intense THz pulse, the bonds inside the metamolecule (via the bridges) become weakened quite rapidly, causing the metamolecule to split into two metaatoms. This abrupt temporal change, also referred to as the time interface, results in the spectral conversion of the incoming THz pulse.

The results presented in the manuscript, while showing potential, have some parallels with previously published work [Ref. 37]. The manuscript differentiates itself through the introduction of time-varying elements (in this case, superconductors), the choice of THz frequency for time-varying excitation of the platform, and the implementation of electrical control on the spectral conversion.

Reply: We are very grateful to the referee for the positive comments on our work. We have addressed all the comments raised by the reviewer in the replies below. We hope the reviewer can recognize us for our efforts to improve the work.

However, it would be beneficial if the authors could elaborate further on the advantages of their approach. Offering additional context, implications, and potential impacts of their platform could improve the presentation of the manuscript.

Reply: Thank the reviewer for the constructive suggestion on our work. This work provides a new platform for achieving linear frequency conversion using a time-varying metasurface. The distinct advantages of our approach are as follows:

Firstly, we achieve linear frequency conversion by rapidly altering the coupling interaction (or “breaking the bond”) between two metallic resonators, which is an entirely different principle. The intense THz pump can trigger transitions of the superconducting film within about 1 ps, comparable to the oscillation period of the THz

probe pulse. Along with the drastic change in the impedance of the superconducting microbridge during the phase transition, the transmission spectrum changes significantly. As a result, the conversion efficiency of the time-varying metasurface we designed is more than two orders higher than that in Ref. 37. Moreover, the conversion efficiency in our work can be adjusted by varying the pump-probe delay and applied voltage, which has significance for advancing the application of the time-varying metasurface.

Secondly, our developed theoretical model explains the experimental results in the time-varying process well. In this model based on the coupled-mode theory, a complex coupling coefficient as a function of time is introduced. It provides a clear physical concept for linear frequency conversion. The results calculated by our model are in good agreement with the experimental results.

We have accepted the suggestions and made the following supplement in the revised manuscript and supplementary information on the context, implication, and potential impacts of the platform.

(1) The context of the platform

In the second paragraph, we added or revised the following sentences concerning the need for time-varying platforms to study time-modulated metasurface. *“At microwave frequencies, the periodically time-modulated metasurfaces have exhibited application prospects in wireless communication, radar, and other fields^{37,38}. Despite the rapid progress, developing the terahertz (THz) time-modulation devices still faces tremendous challenges since the modulation speed of electronic switches limits the extent to which the input signal frequency can be shifted.”*

(2) The implication of the platform

We have completely revised the theoretical model part. We have presented a detailed explanation of physics related to each part of the Hamiltonian in Eq. 1 and added the partial derivation equations for the resonator modes, including the losses. The refined model is much clearer and includes more details. Please refer to the “Time-varying coupled-mode model” section in the revised manuscript.

We have calculated the conversion efficiency using an integration method. We only integrated the frequency components beyond the frequency spectrum of the input pulses and calculated the ratio of the converted power to the total input power. The conversion efficiency still exceeded 4%. The calculated conversion efficiency from the experimental and theoretical data is consistent. It further illustrates that we have achieved efficient frequency conversion. Please refer to Supplementary Note 7 in the revised supplementary information.

We have updated Fig. 3d and refined the discussion on phase coherence in the frequency conversion process in the revised manuscript. The original Fig. 3d is inaccurate in illustrating the coherence between the converted and input waves. In the original figure, we fixed the frequency of the converted wave at 0.47 THz. However, the peak frequency of the converted wave varies with t_{pp} . In the revised Fig. 3d, we plot the measured and calculated relative phase of the converted wave at the peak frequencies with respect to the phase of the incident wave at the center frequency. After correcting the method, we found that the theoretical and experimental results conformed better. We believe the refined results better illustrate the coherence of the frequency conversion process. Please refer to Fig. 3d and the relevant explanation in the “Coherent frequency conversion in time-varying meta-molecules” section of the revised manuscript.

(3) The potential impacts of the platform

A potential impact of the proposed platform is offering a powerful tool to study fundamental physics, such as topological phase transition. One of the main reasons for using time-varying platforms is essential in topological photonics research is the ability to dynamically induce and manipulate topological phase transition. In static photonic systems, the topological properties are fixed, limiting our ability to explore the rich dynamics associated with topological phase transitions. By introducing time-dependent perturbations, we can dynamically modulate the photonic structure, leading to the emergence of new topological phases and exotic phenomena. The photonic devices with tunable topological characteristics could be designed, which can be leveraged for a wide range of applications, from robust optical communication systems to efficient quantum

information processing. We have added the envisioned application of topological photonics in the revised Supplementary Note 13. In our research, we utilize superconducting microbridges to couple meta-atoms, enabling us to dynamically adjust the coupling strength (J) with intense THz pulses at the picosecond level. This platform allows us to explore the temporal dynamics of 1D topological chains and 2D topological metasurfaces.

Another potential impact of the platform is contributing to the development of novel photonic devices. The proposed time-varying metasurface provides a method for efficient and controllable frequency conversion. Though the present experimental setup is complicated, we can optimize the design to reduce the size and energy consumption of the device. Furthermore, the design methodology and the theoretical model developed in this work can be extended to other materials and devices with time-varying properties. In the revised Supplementary Note 14, we have provided a design of an on-chip frequency converter.

Reviewer #3

The authors propose a method of frequency conversion in a time modulated meta-surface, where two split-ring resonators are connected with superconducting bridges. A pump pulse allows controlling the superconducting element, which can be either inductive or purely resistive, depending on the temperature rise induced by the pulse. As a result, the electromagnetic resonance of the system is altered switching between two configurations: coupled resonator case, with two different frequencies ω_1 and ω_2 and uncoupled resonator case with a single frequency ω_0 . The authors perform pump-probe experiments where they study the dynamics of the switching regime, and interpret their result in terms of coherent frequency conversion.

This work clearly bears strong inspiration from a prior work by Lee and al. that was published in Nature Photonics, <https://doi.org/10.1038/s41566-018-0259-4> (ref. 37): a fact acknowledged by the authors. In particular, their explication of the frequency conversion effect resembles strongly to the concept of “temporal boundary” introduced in Ref. 37. Nevertheless, the meta-surface device proposed by the authors operates on a quite different principle. Also, the experimental data and the corresponding modelling seem to be much more clear than the ones in Ref. 37. Indeed, in the present case there is a clear pic that corresponds to the frequency converted signal, whereas in the data of Ref. 37 the spectral features were fairly broad (Fig. 3a in Ref. 37). The paper thus seems to contain sufficient novelty and could, in principle, be published in Nature Communications, however it still lacks sufficient details, clarity, and even scientific insight. Some (including major) corrections are needed:

Reply: We are very grateful to the referee for the positive comments on our work and the comprehensive summary of our work. We have addressed all the comments raised by the reviewer in the replies below. We hope the reviewer can recognize us for our efforts to improve the work.

1. It will be very helpful for the reader if the authors indicate clearly the “original frequency” and the “converted frequency”, f_c , in Figures such as 2(c,d,e, f). Is the

original frequency the one 0.34 THz?

Reply: We sincerely thank the reviewer for this valuable suggestion. We have accepted the advice of the reviewer and updated Fig. 2c-f in the revised manuscript. The updated figure is shown in Fig. R9 as well.

In our understanding, “the original frequency” refers to the center frequency of the input signal, *i.e.*, 0.34 THz. It is indicated by the blue arrow in the updated Fig. 2c, d. The red dashed lines in the updated Fig. 2c, d show the converted frequency regions. The peak frequency of the input signal is 0.34 THz (approaching f_1). It is indicated by the blue dashed lines in the updated Fig. 2e, f. The peak frequencies of converted waves (f_c) are indicated by the red dashed line in the updated Fig. 2e, f.

Fig. R9 (updated Fig.2) (a) Microscopic image of the fabricated superconductor-metal hybrid meta-molecule array. (b) Normalized reference power spectra obtained after the input pulse, centered at 0.34 THz, transmitting through a magnesium oxide (MgO) substrate. Two-dimensional plots of the measured (c) and calculated (d) output power spectra as a function of the pump-probe delay (t_{pp}). The red marker curves indicate the converted frequency region. The blue arrows indicate the center frequency of the input pulse. The black dashed lines indicate the moments when the frequency conversion occurs and ends. (e) Measured and (f) calculated output power spectra for various t_{pp} (each curve is vertically offset by 0.15). The red dashed lines indicate the frequency shift of the converted wave peak. The blue dashed lines indicate the center frequency of the input signal.

To fully address this comment, we have added the following sentences in the second paragraph after Fig. 2 in the revised manuscript. *“The red marker curves indicate the converted frequency region. The blue arrows indicate the center frequency of the input pulse. The black dashed lines indicate the moments when the frequency conversion occurs and ends.”*

2. How was the converted power in Fig. 3b estimated: is it the peak power at the frequency f_c (dots in Fig. 3a), or some integration was involved?

Reply: We sincerely thank the reviewer for pointing out the vague statement. The converted power in Fig. 3b was estimated using the peak power at the converted frequency, represented by the dots in Fig. 3a. No integration is involved in the calculation. The calculation method is consistent with the original Ref. 37. We follow the same approach as the original Ref. 37 to verify whether the experimental results satisfy the linear frequency conversion relationship. We fitted the curve of the peak power of the converted wave as a function of the input power, demonstrating the linear relationship in the frequency conversion. We also utilized an integration approach to calculate the conversion efficiency in the revised supplementary information. Please

see Supplementary Note 7 for details.

Fig. R10 (updated Fig.3a, b) (a) Power spectra obtained under different incident power when $t_{\text{pp}} = -3.7$ ps. (b) Linear relationship between the converted power and the input power at 0.47 THz. The measured data are depicted with solid circles, corresponding to the points in **a**. The dashed line represents the fitted data.

To fully address this comment, we have added the following sentence in the first paragraph after Fig. 3 in the revised manuscript. “*We analyzed the converted peak power at 0.47 THz (indicated as “A-E” in Fig. 3a) as a function of the input power.*”

3. In the introduction, the authors claim a possibility of applications, however this claim seems overstatement as their device is operated at 4K.

Reply: We sincerely thank the reviewer for this insightful comment. Although the proposed superconducting-metal hybrid structure in our work requires an operating temperature of 4 K, one potential application of this work is providing a powerful platform for studying Floquet physics and topological photonics, which is rarely accessible in previous studies. For example, we envision the proposed metasurface in which modulation dynamics and their interplay with nontrivial material properties, such as non-Hermiticity and nonreciprocity, will introduce a novel approach to wave

manipulation in the temporal period.

In addition, the low-temperature instruments have developed rapidly in recent years. These advancements have led to the miniaturization of 4 K refrigeration systems. We are optimistic that the more compact and efficient cryogenic solutions will soon unlock a broader range of applications for low-temperature THz devices.

More importantly, the design methodology and the theoretical model developed in this work also apply to other time-varying devices based on materials with ultrafast responses at room temperature. For example, it has been reported that some materials, such as graphene and semiconductor quantum dots, have ultrafast response times under the optical pump or the intense THz pump. Moreover, as technology progresses and new materials with ultrafast response are found, the applications of the time-varying devices will become more and more promising.

4. The authors talk about “coherent frequency conversion”, and they present data and simulation on the phase difference in Figure 3d. In the simulation, there is a strong 2π -jump of the phase, indicating much stronger variations than in the experiment. Also, the experimental phase at -8ps is almost zero, while the simulation predicts a value of almost $\pi/4$, which is a significant difference. The trends for the simulated and measured phase is fairly different. The claim that the authors observe a coherent effects is thus strongly overrated, as the experimental phase behaves very differently from the model; that claim should be removed from the title and the introduction.

Reply: We sincerely thank the reviewer for drawing our attention to the phase shifts in the frequency conversion effect. We have found some inaccuracies in the previous analysis of the relative phase in Fig. 3d, and we have corrected them in the revised manuscript.

In the original Fig. 3d, we first calculated the phase difference between the output and input waves at 0.47 THz for different t_{pp} , and obtained the relative phase ($\Delta\phi$) by subtracting the reference phase at $t_{pp} = 0$ ps. A strong 2π -jump of the phase in the simulated data is due to the phase wrap. We reanalyzed the data and concluded that the

original Fig. 3d is inaccurate in illustrating the phase coherence between the converted and input waves. Since the peak frequency of the converted wave varies with t_{pp} , as shown in Fig. R11a, it means that there is no converted peak at 0.47 THz for some t_{pp} . Furthermore, the frequency conversion phenomenon almost disappears when $t_{pp} \geq -1.5$ ps, the calculated data after $t_{pp} \geq -1.5$ ps in original Fig. 3d should be removed.

Fig. R11(updated Fig. 3c, d) (a) Frequencies of the converted wave peaks (f_c) as a function of t_{pp} . The measured f_c is denoted by open circles, while the solid curve represents the calculated f_c . (b) The relative phase ($\Delta\phi$) of the converted wave at the peak frequencies (corresponding to Fig. R3c) with respect to the phase of the input wave at the center frequency. The open circles show the measured $\Delta\phi$, and the solid curve shows the calculated $\Delta\phi$.

In the revised Fig. 3d, we show the measured and calculated $\Delta\phi$ of the converted wave at the peak frequencies (corresponding to Fig. R11a) with respect to the phase of the incident wave at the center frequency. The updated Fig. 3d is copied in Fig. R11b. The phase of the converted wave at peak frequencies can be controlled by t_{pp} , and the measured $\Delta\phi$ is stable. This phase stability implies a certain coherence between the input and output signals. Though it is usually challenging to accurately measure the phase coherence experimentally, the converted wave clearly preserves the phase information of the input wave. Thus, the frequency conversion is coherent. The prediction of the theoretical model based on the coherent interaction is in good

agreement with the experimental results, also strongly confirming the coherent frequency conversion.

After correcting the method of evaluating the measured and calculated $\Delta\phi$ between the converted and the input waves, it was found that the theoretical and experimental results conformed well. Therefore, we maintain the claim of coherence in the title and abstract.

To fully address this nice comment, we have made the following revisions:

(1) We have replaced Fig. 3d with Fig. R11b and amended the caption of Fig. 3d in the revised manuscript. *“(d) The relative phase ($\Delta\phi$) of the converted wave at the peak frequencies (corresponding to Fig. 3a) with respect to the phase of the input wave at the center frequency. The open circles show the measured $\Delta\phi$, and the solid curve shows the calculated $\Delta\phi$.”*

(2) We have rewritten the sentences in the second paragraph after Fig. 3 in the revised manuscript. *“We also conducted an analysis of the relative phase ($\Delta\phi$) of the converted wave at the peak frequencies (corresponding to **Fig. 3c**) with respect to the input wave at the center frequency. The measured and calculated $\Delta\phi$ as a function of t_{pp} are shown in **Fig. 3d**. The measured $\Delta\phi$ roughly exhibits a linear increase from nearly 0 to 1.7π as t_{pp} increases from -8 ps to -1.7 ps. Although accurately measuring the phase coherence in the frequency conversion process is experimentally challenging, the converted field still clearly retains phase information of the input field. Thus, the frequency conversion is coherent. The prediction of the theoretical model based on the coherent interaction is in good agreement with the experimental results, also strongly confirming the coherent frequency conversion.”*

5. The electrical measurements in Figure 4 present an interesting add-on but, besides the description of the device and a few measurements, this configuration is not very well exploited and not sufficiently commented. This is clearly a means for a static control of the meta-surface, the physical effects being rather trivial (killing the superconductivity by Joule heating), but what is the point of that? What does this static mechanism has to do with the time-varying modulation and what do we learn from this?

This part can be removed; unless an experimental demonstration of a new effects (i.e. applying an RF bias to periodically modulate the coupling constant by all electrical means).

Reply: We sincerely thank the reviewer for this valuable comment. We have accepted the reviewer's suggestion and moved this part from the main article to the supplementary information.

As the reviewer mentions, the physical mechanism of electrical control in Fig. 4 is that electrical heating suppresses the superconducting state of microbridges. Though the static electrical control does not introduce novel physical effects, it provides a convenient means to tune the “bond.” In other words, the conversion efficiency can be tuned by voltage bias. Compared to previous work, it offers a more accessible method to control the output of converted waves.

On the other hand, we agree with the referee that the periodic modulation of the coupling constant by all electrical means will be a promising direction for THz time-varying devices. At microwave frequencies, a variety of time-domain modulated metasurfaces has been developed and shown many intriguing phenomena (*Nat. Commun.* 9, 4334 (2018), *Nat. Electron.* 4, 218 (2021)). Moreover, they have good application prospects in wireless communication, radar, and other fields. However, due to the lack of materials capable of ultrafast modulation at THz frequencies and limitations in the frequency resolution of THz time-domain spectroscopy, the RF-modulated THz time-varying metasurfaces are still challenging.

To fully address this comment, we have amended the fourth paragraph after Fig.3 in the revised manuscript. “*We experimentally observed the tuning of the efficiency of conversion peak with the applied voltage, which can be attributed to the electric heating effect (see Supplementary Note 10 for details). The electric tuning offers a more accessible method to control the output of the converted waves, which has significance for the application of the time-varying metasurface. We also found that the loss of metasurface is a crucial factor affecting the conversion efficiency (see Supplementary Notes 10-12 for details).*”

6_1. The model contained in Eq. (1) is pretty interesting, and seems to explain pretty well the data, but it has not be detailed enough, not even in the supplementary materials. It will be very interesting if the authors could provide more details and developments based on the Hamiltonian (1), and probably provide analytical expressions for the frequency and phase shift in certain simplified cases.

Reply: We sincerely thank the reviewer for this valuable comment. In the revised version, we have completely revised the theoretical model part. We have presented a detailed explanation of physics related to each part of the Hamiltonian in Eq. 1 and added the partial derivation equations for the resonator modes, including the losses. In the revised manuscript, the model is much clearer and includes more details. We also have corrected the typos and mistakes in the Hamiltonian.

According to the Hamiltonian and the mechanism explaining the frequency conversion process in Fig. 1c, the frequency shift is $\Delta f = f_c - f_{in} = |J_{max}| - |J_c|$. The frequency shift as a function of the pump-probe delay roughly takes the vertical inversion of the temporal transition of $J(t)$. It is confirmed by our experimental observation (see Fig. 3c).

It is difficult to provide an analytical expression for the phase shift because the coupling of the system is time-dependent, and the spectrum of the probe field is broad. The phase shift between the converted and input fields originates from two processes. First, the excitation process induces a phase shift, and the phase of the excited supermode can be different from the input field. Second, once the meta-molecule is excited and then broken into two separate meta-atoms, the coupling changes during radiation. This process accumulates a phase shift to the converted field. Because of this complicated process, we cannot derive an analytical expression for the phase shift. However, the numerical simulation of our model reproduces the experimental observation of the phase shift, as shown in the updated Fig. 3d of the revised manuscript.

To fully address this nice comment, we have made the following revisions:

(1) We have added the following sentence in the first paragraph after Fig. 1 in the

revised manuscript. “Basically, the meta-molecule consists of two “bonded” meta-atoms, which are two mirror-symmetric planar metallic resonators connected via two superconducting microbridges.”

(2) We have added a paragraph behind the first paragraph after Fig. 1 in the revised manuscript. “In the experiment, the meta-molecule is formed by two bonded and mirror-symmetric meta-atoms. Thus, we can simply consider that the two meta-atoms are driven by a THz probe pulse of $E_p(t)$ with the same phase and strength (κ_e). The two meta-atoms have the resonance frequencies of f_a and f_b , and the total loss rates of κ_a and κ_b . We use the annihilation operators of a and b for the two resonator modes. The meta-atoms are coupled via the microbridges with a strength of $J(t)$. A pump pulse is used to tune this coupling strength temporally. During the time-varying process, the superconductivity of the microbridges is suppressed rapidly, and they are switched to the normal state, leading to an increased Ohmic loss to the meta-atoms. We introduce a time-varying loss rate of $\kappa_j(t)$ to the meta-atoms for modeling the effect of varying coupling on the system loss.”

(3) We have amended Eq. 1 and added the following sentences in the original second paragraph after Fig.1 in the revised manuscript. “The Hamiltonian for a typical system comprising two coupled meta-atoms can be expressed as^{48,49}:

$$\begin{aligned}
 H = & f_a a^\dagger a + f_b b^\dagger b + [J(t)a^\dagger b + J^*(t)ab^\dagger] \\
 & + i\sqrt{2\kappa_e}E_p(t)(a^\dagger + a) \\
 & + i\sqrt{2\kappa_e}E_p(t)(b^\dagger + b),
 \end{aligned} \tag{1}$$

The first two terms describe the free energy of the meta-atoms. The third term indicates the time-varying coupling between two meta-atoms. The last two terms are for the driving of the two meta-atoms. The decay of the meta-atoms is included in the model using the Lindblad operators. As well known, the spectral response is highly sensitive to the bond or coupling strength. By incorporating a time-dependent bond into the meta-molecule, we can effectively model frequency conversion in the transition region. According to the Hamiltonian in Eq. 1 and the mechanism shown in **Fig. 1c**, the frequency shift between the converted and the input field is roughly given by

$$\Delta f_c = |J|_{\max} - |J(t_j)| \tag{2}$$

The phase shift between the two fields includes two contributions: (i) the phase difference when the supermode of the meta-molecule is excited; (ii) the phase accumulating during the time-dependent radiation when the coupling varies. Thus, it is difficult to provide an analytical estimation. The phase of the converted field will be experimentally measured and numerically calculated.”

(4) We have added a paragraph behind the original second paragraph after Fig. 1 in the revised manuscript. “We take into account the decay of the meta-atoms with the Lindblad operators. For an arbitrary operator of Q , we use^{48, 49}

$$\frac{\partial Q}{\partial t} = i [H, Q] + L\{\kappa_a + \kappa_j(t), a\}Q + L\{\kappa_b + \kappa_j(t), b\}, \quad (3)$$

where the Lindblad operator takes the form $L\{\kappa, O\}Q = \kappa(2 O^\dagger Q O - O^\dagger O Q - Q O^\dagger O)$, and replace the operators with their mean values: $\alpha = \langle a \rangle$ and $\beta = \langle b \rangle$. Then, we obtain the Langevin equations as follows,

$$\begin{aligned} \dot{\alpha} &= -[i f_a + \kappa_a + \kappa_j(t)]\alpha(t) - i J(t)\beta(t) + \sqrt{2\kappa_e} E_p(t), \\ \dot{\beta} &= -[i f_b + \kappa_b + \kappa_j(t)]\beta(t) - i J^*(t)\alpha(t) + \sqrt{2\kappa_e} E_p(t). \end{aligned} \quad (4)$$

When the coupling is strong, the real part of J is much larger than its imaginary part. We can study the dynamics of the eigen supermodes of the two meta-atoms, to a good approximation, on the basis of $(a \pm ib)/\sqrt{2}$. For $f_a = f_b = f_0$, the eigen frequencies are $f_0 + |J(t)|$ and $f_0 - |J(t)|$. These supermodes can be driven with the same strength. To selectively excite one supermode, we tune the probe field to resonate with it.”

(5) We have amended or added the following sentences in the original third paragraph after Fig. 1 in the revised manuscript. “where τ_J is the duration for breaking the bond of the meta-molecules, $X_m = 0.167$ THz and $R_m = 0.03$ THz. Here, we set $\tau_J = 1.5\pi$ ps. The time evolution of J in the calculation is shown in **Fig. 1c**. Before the bond breaks, we have $|J|_{\max} = 0.167$ THz. After the bond breaks, the coupling is switched off, becoming $|J|_{\min} = 0.03$ THz. The additional Ohmic loss can be assumed to be

$$\kappa_j(t) = \kappa_0 [1 - sw(t, t_J, \tau_J)] \quad (7)$$

with $\kappa_0 \approx 0.02$ THz. Before the bond breaks, the loss is zero. It becomes κ_0 after the bond is broken by the pump pulse.”

(6) We have amended the original Eq. (4) in the revised manuscript.

“
$$E_p(t) = A\mathcal{E}(t) \cos[f_{in}(t - t_p)] \quad (8)$$
”

6_2. Also the physics contained in the Hamiltonian is similar to the one used to describe the dynamical Casimir effect

(see <https://journals.aps.org/prl/abstract/10.1103/PhysRevLett.98.103602>), the authors could comment their work within this context.

Reply: We sincerely thank the reviewer for pointing out the relevance of the literature about the dynamical Casimir effect and our work. We have added the recommended paper to the reference list in the revised manuscript.

The literature (*Phys. Rev. Lett.* 98, 103602 (2007), *Phys. Rev. B* 72, 115303 (2005)) develop a fundamental theory for explaining the quantum vacuum radiation from a time-varying cavity QED system, known as the so-called dynamical Casimir effect. In the configuration for this dynamical Casimir effect, the two modes, which can be two cavity modes or polariton modes, are coupled in the ultrastrong coupling regime. Thus, the antiresonant terms of the light-matter interaction play an important role. In the ultrastrong coupling regime, the vacuum is squeezed due to the antiresonant terms and is driven to the ground state, leading to the excitation of the cavity or polariton mode as a squeezed quantum ground state from the vacuum state. When the system is temporally modulated, in other words, the system “boundary” is time-varying. These photons emit from the cavity as quantum vacuum radiation. This developed theory can be extended to an electromagnetic system with time-varying boundaries. However, the antiresonant terms must play an essential role in radiation, and the quantum vacuum state must include photons. On the other hand, this radiation is very weak.

In our experiment, we measure the transmission of the THz input pulse and observe a remarkable component at the converted frequency. The electromagnetic system and the probe field are classical. Indeed, the two meta-atoms are strongly coupled and form two supermodes with split frequencies. The classical probe field first

resonantly excites the supermode. After the coupling is switched off, *i.e.*, the bond is broken, the energy of the supermode shifts to a different value. Then, the meta-atom (resonator) radiates the THz wave at the converted frequency. In our theoretical model, the Hamiltonian in Eq. 1 only considers the resonant interaction but excludes the antiresonant coupling. We also use operators for the fields and then replace them with corresponding coherent mean values. In classical treatment, the antiresonant terms indeed would not excite the resonator. As shown in the manuscript, our model can reproduce the experimental results very well. Thus, the radiation from our system originates from the initial excitation of the supermode of the meta-atom rather than from the quantum vacuum radiation caused by the dynamical Casimir effect.

A more precise quantum model may take into account the antiresonant interaction and the squeezing of the quantum vacuum. Quantum vacuum radiation may play some role. However, this radiation will be extremely weak compared to the classical signal. Moreover, a full quantum model is complicated and difficult to calculate, especially for the current case with a strong probe field, including a large number of photons. On the other hand, the classical treatment, without taking into account the dynamical Casimir effect, is sufficient for reproducing the experimental results, as demonstrated in this work. Therefore, we use the simple classical model for interpreting our experimental observations.

To fully address this nice comment, we have added a paragraph after the second paragraph of the discussion section in the revised manuscript. *“The mechanism of THz radiation from our time-varying metasurface is essentially different from the quantum vacuum radiation originated from the dynamical Casimir effect, which has been well interpreted with a quantum theory^{48,49}. In the dynamical Casimir effect, the quantum ground state of a light-matter coupled or two-mode coupled system is initially excited by the ultrastrong coupling via the antiresonant interaction. When the boundary of the system is suddenly changed, the ground state changes and then radiates photons included in its excitation. This radiation is purely a quantum process created from the excited quantum vacuum state. In contrast, the converted field is generated via a classical process in our experiment, although our system is also time-varying. One*

supermode of the meta-molecule is initially excited by the classical THz probe field. Then, the coupling of the meta-atoms is rapidly tuned off. In this case, the energy of the excited supermode shifts and the excited supermode radiates at a new frequency. The radiation from the quantum vacuum state is negligible here.”

7_1. It will be also helpful if the authors could comment on the selection rules for their structures. For instance, because of the symmetry of their meta-atom, the uncoupled mode should be degenerate into two modes, a dark mode, where the currents in the inductive loops oscillate in phase opposition, and a bright mode, where they oscillate in phase. Also, similar rules should apply for the coupled system, where one of the coupled modes should be much brighter than the other.

Reply: We sincerely thank the reviewer for this valuable suggestion, which has helped us improve our manuscript significantly. When the two meta-atoms are decoupled, the eigen supermodes can be the symmetric and antisymmetric modes, *i.e.*, the bright and dark modes, corresponding to $(a + b)/\sqrt{2}$ and $(a - b)/\sqrt{2}$. They are degenerate in frequency. In this supermode, the currents in the dark (bright) mode oscillate in the opposite (the same) phase. The THz field incidents to the planar meta-molecule can only excite the bright supermode. Without coupling, the eigen supermodes can also be chosen as $(a + i b)/\sqrt{2}$ and $(a - i b)/\sqrt{2}$. Below, we explain that we need to take the second basis when the two meta-atoms are bonded.

In our experiment, the two meta-atoms are mirror symmetric. The key point of our setup is that the coupling of the two meta-atoms is different from the conventional coupled resonators because the coupling strength is a complex number. For simplicity, we consider the two uncoupled meta-atoms to have the same resonance/transition frequency f_0 . The real part of J is negligible. Thus, we can view J as a purely imaginary number. In this case, the eigen supermodes are $(a + i b)/\sqrt{2}$ and $(a - i b)/\sqrt{2}$, with eigenfrequencies shifted by $|J|$ and $-|J|$, respectively. When the probe field incident to the meta-atoms with the same phase as in our experimental setup, these two supermodes

are excited with the same strength. However, we can selectively excite the supermodes by tuning the probe frequency. Our experimental observation confirms this. Please see the third paragraph after Fig.3 and Supplementary Notes 8 and 9 for details.

7_2. This means that not only the coupling strength J is modulated, but also the radiation loss (parameters k_a and k_b in Eq.(1)). Have the authors taken into account this effect, that can be quite significant, and how does it affect the data modelling?

Reply: We sincerely thank the reviewer for this valuable comment. In our model, k_a and k_b describe the total loss of the two resonators when the coupling is on, and J is a purely imaginary number. They include radiation and Ohmic loss. The radiation loss originates from the THz radiation from the resonators to free space, which is determined by the geometry of the resonators. The Ohmic loss is caused by the energy loss of electrical resistance.

In our experiment, the two meta-atoms are two metallic resonators. Their radiation and Ohmic loss remain unchanged with the variation of J . The two meta-atoms are coupled via two superconductive microbridges initially. Before the arrival of the pump pulse, the microbridges are superconducting. Thus, the induced loss to the resonators/meta-atoms is negligible. After the microbridges are tuned to the normal state, the resistance of the microbridges may cause remarkable Ohmic loss. Modulation of J mainly changes the resonance frequency of the supermodes and maybe slightly changes the Ohmic loss. By applying an additional identical decay rate $\kappa_j(t)$ to the resonators, we indeed consider the effects of the modulation of $J(t)$ on the loss. Note that we model the effect as a total loss change. In the previous manuscript, we already included this time-dependent decay but did not provide a detailed explanation and description for the model. In the revised version, we have clarified this time-varying loss. Our theoretical model can well reproduce the experimental observation, confirming its validity.

The change of J does not modify the radiation capability of the resonators. Nevertheless, tuning the microbridges from the superconducting to the normal state has

small effects on the Ohmic loss. Thus, the total loss of the system may change slightly when the coupling is modulated. It is confirmed by the numerically simulated spectral profiles with and without coupling, as shown in Fig. 1 of the main article.

8. The authors should definitely provide field maps illustrating the resonant modes.

Reply: We sincerely thank the reviewer for this constructive suggestion. We simulated the electric field distribution of three resonant modes, as shown in Fig. R11 below. Please refer to Fig. S1d in the revised supplementary information.

Fig. R11. Simulated electric field distribution at the resonance frequencies of f_1 , f_2 , and f_0 , corresponding to CTP, SBDP, and BDP mode.

Figure R11 displays the simulated electric field distribution of three resonant modes before and after “breaking the bond.” In the superconducting state, the microbridges function as conductive channels, which allow the charges to flow through. In that case, the two resonators are strongly coupled. The resonance mode splitting generates two resonant dips in the transmission spectrum, which correspond to f_1 and f_2 . At f_1 , the positive and negative charges accumulate in the upper and lower rings, respectively. Correspondingly, an electric dipole resonance is induced, similar to the charge transfer plasmonic (CTP) mode in plasmonic dimers. At f_2 , the positive and negative charges are accumulated in the upper and lower part of each resonator, leading to a screened bonding dimer plasmonic (SBDP) mode with a shielded electric field in

the gap.

In the normal state, the losses in the NbN microbridge significantly suppress the conductive coupling interaction. The two resonators are weakly coupled, and there is only one resonant dip (corresponding to f_0) in the transmission spectrum. At f_0 , the highest charge density appears at the upper and lower ends of the unit cell. It results in a hybrid mode combining the two electric dipoles, which is akin to the bonding dimer plasmonic (BDP) mode observed in plasmonic dimers.

9. The overreach of this work should be expanded, for instance what type of effects related in the topological dynamics and Floquet physics can be expected? How shall the structure designed by the authors be modified in order to make such effects observable?

Reply: We sincerely thank the reviewer for the valuable suggestions and comments. In our work, two meta-atoms are coupled by superconducting microbridges. The coupling strength (J) can be temporally tuned by an intense THz pulse at the picosecond level. By extending the design of two coupling meta-atoms to a 1D topological chain shown in Fig. R12 or a 2D topological metasurface shown in Fig. R13, and fast tuning the coupling strength, we have the capability of studying the temporal dynamics using the topological metasurface.

Fig. R12. A 1D chain of coupled meta-atoms for studying the topological dynamics of the SSH model.

The configuration shown in Fig. R12 creates a 1D chain of meta-atoms. We name the two coupled meta-atoms as A and B, respectively. In each unit cell, the meta-atoms of A and B are coupled with a rate of J_1 . The strength of the A-B coupling between the unit cells is J_2 . Then, the meta-atoms form a standard Su-Schrieffer-Heeger (SSH) model. Although the topology of the SSH model has been investigated in the metasurface platforms, the topology-related temporal dynamics in the THz regime are rarely reported.

Fig. R13. A 2D metasurface consisting of coupled meta-atoms for studying the topological dynamics of 2D topological THz photonics.

The configuration in Fig. R13 provides a possible design for a 2D topological metasurface (Ref. 50). The topology feature is crucially dependent on the three coupling rates: J_1 , J_2 , and J_3 , where J_3 denotes the coupling between the upper and lower meta-atoms. We can change the coupling ratios in time by fast tuning the coupling rates. Then, the temporal transition between topological phases with different Chern numbers and the dynamic emergence of the edge and corner states can be investigated in the two configurations.

In this work, the two meta-atoms are coupled via the superconducting microbridges. This coupling can be switched off within a few picoseconds with a THz

pulse and restored quickly. Thus, by applying a series of THz pulses with a temporal period (T), we can periodically modulate meta-atom couplings at a specific frequency ($1/T$). It means that we create a periodically modulated system. By doing so, we create Floquet metamaterials and introduce a “synthetic” dimension to the system. Hence, we can study Floquet physics, such as the formation of Wannier-Stark ladders (*eLight* 2, 8 (2022)). If we periodically modulate the coupling in the topological metamaterials, as shown in Figs. R12, R13, then we can study and then reveal topological Floquet physics.

To fully address the valuable comment, we have added Figs. R12, R13, and the above discussion in Supplementary Note 13. Please refer to the revised supplementary information.

We have also added the following sentences in the second paragraph of the discussion section in the revised manuscript. *“If we periodically modulate the coupling in the topological metamaterials with a train of pumping pulses, we can study topological Floquet physics. We envisioned the extended research work in topological dynamics and Floquet physics based on the proposed metasurface (see Supplementary Note 13 for details).”*

Reviewer #2 (Remarks to the Author):

In the revised manuscript, the authors have thoughtfully addressed the issues previously raised. However, for this work to be considered for publication in Nature Communications, I suggest that the following points could benefit from additional clarification and emphasis:

The authors present their metasurface structure as a novel innovation. While this approach is indeed creative, the claim of novelty merits some further discussion. The concept of bond breaking, as introduced here, shares similarities with earlier approaches where two resonances merge into one at the temporal boundary. Previously, this was described as "meta-atom merging," and in the current work, it is labeled "bond-breaking." The steady-state spectra in both instances exhibit a similar merging of two resonances into one. The main distinctions seem to lie in the use of a superconducting bridge (rather than semi-insulating GaAs) and slight variations in meta-atom structures. These differences, while notable, may not constitute a fundamentally new principle, particularly given the analogous use of THz pulses for bond-breaking in this study and optical pump pulses for meta-atom merging in prior research.

The proposal for a topological device is quite intriguing. However, it should be noted that a concept similar to the other proposed application has already been discussed, though with certain differences in implementation, as detailed in the work of F. Miyamaru et al. (Ultrafast Frequency-Shift Dynamics at Temporal Boundary Induced by Structural-Dispersion Switching of Waveguides, Phys. Rev. Lett. 127, 053902, 2021).

In the discussion of efficiency, it would be advantageous for the authors to also address the relative shift of the new frequency content compared to the original. Prior studies suggest that efficiency improves as the relative frequency shift is reduced. Thus, a discussion on efficiency that does not include specifics about the frequency shift may not adequately reflect the underlying dynamics.

The manuscript's claim to be the first in demonstrating 'coherent' frequency conversion via a temporal boundary is misleading and inaccurate. This aspect, particularly the importance of relative phases, has been comprehensively addressed in earlier work, indicating that a revision or further clarification in the manuscript might be necessary.

Furthermore, a recent study that utilized a temporal boundary in a Fabry-Perot cavity to enhance conversion efficiency, achieving up to 33% (significantly higher than what is reported in this manuscript), should be considered. This study (K. Lee et al., Resonance-enhanced spectral funneling in Fabry-Perot resonators with a temporal boundary mirror, Nanophotonics, 2022) achieved a frequency shift of approximately ~ 0.2 THz from an original peak frequency of around 0.5 THz, with the newly generated frequencies distinctly separated from the input spectrum. This work is pertinent to the claims made in the manuscript and deserves attention.

In conclusion, I believe that addressing these points with appropriate depth and context would significantly enhance the manuscript. Without these clarifications, it would be difficult for me to recommend this work for publication in Nature Communications.

Reviewer #3 (Remarks to the Author):

The authors have made significant effort to address all the reviewers' comments and have provided corresponding changes to the article main text and the supplementary material. Especially, very good agreement between experiment in model is now achieved with Figure 3d, which shows the phase relation between the input and converted signal. The scientific quality of the newer version is thus strongly improved, and I have no further objectives to publications.

Dear Editor and Reviewers,

We would like to thank you for your efforts on our manuscripts. We greatly appreciate the constructive and insightful comments from the Editor and Reviewers. In this letter, we have carefully addressed the comments and queries raised by Reviewer 2 (comments from the referees are in blue italicized). Point-by-point responses to the referee are listed below. Besides that, we attach an additional manuscript in which the major text changes are marked in red. We believe that the revised manuscript has taken full account of the points raised by the reviewer, and we hope you will be satisfied with the revised version of this paper.

Best regards,

The authors

Reviewer 2

In the revised manuscript, the authors have thoughtfully addressed the issues previously raised. However, for this work to be considered for publication in Nature Communications, I suggest that the following points could benefit from additional clarification and emphasis:

-1. The authors present their metasurface structure as a novel innovation. While this approach is indeed creative, the claim of novelty merits some further discussion. The concept of bond breaking, as introduced here, shares similarities with earlier approaches where two resonances merge into one at the temporal boundary. Previously, this was described as "meta-atom merging," and in the current work, it is labeled "bond-breaking." The steady-state spectra in both instances exhibit a similar merging of two resonances into one. The main distinctions seem to lie in the use of a superconducting bridge (rather than semi-insulating GaAs) and slight variations in meta-atom structures. These differences, while notable, may not constitute a fundamentally new principle, particularly given the analogous use of THz pulses for bond-breaking in this study and optical pump pulses for meta-atom merging in prior research.

Fig. R1 Schematic diagram of the “meta-atoms merging” and the “bond-breaking” process at the temporal boundary.

Reply: We deeply appreciate the recognition of our work as a creative approach and the constructive comment. The concept of merging meta-atoms upon ultrafast optical excitation has been introduced in our present and revised versions. Indeed, linear frequency conversion is for the first time theoretically proposed and experimentally demonstrated in K. Lee's work (*Nat. Photonics* 12, 765(2018)) by using the mechanism of merging an artificial molecule consisting of two meta-atoms with different size into

a single bigger meta-atom via ultrafast optical excitation. However, due to the significant loss of the GaAs substrate, the merged mode has a broad resonance profile and a low Q factor, leading to low conversion efficiency.

In contrast to the K. Lee's work, our work uses essentially different mechanism for linear and phase controllable frequency conversion. The schematic comparison of the “meta-atoms merging” process in the K. Lee's work and the “bond-breaking” process in our work are shown in Fig. R1. In the previous work, two coupled smaller meta-atoms suddenly merge into a single bigger one at the temporal boundary. In our approach, we introduce a design featuring a superconducting (SC) microbridge to “bond” two metallic meta-atoms. The ultrafast THz pulse makes the microbridge transit from the SC to normal states. Thus, the bond between the two meta-atoms is broken. In this, the coupling strength between the resonators is temporally modulated. The resonance profile is clean and the Q factor is high before and after the temporal boundary, resulting in a high conversion efficiency. The two meta-atoms remain their properties (keep the sizes and thus the resonance frequencies) and their bond breaks after the temporal boundary, which makes the physical picture clean and easily understandable.

In addition, the temporal modulation of the coupling strength could be used to study many new physical phenomena, such as topological phenomena and non-Hermitian physics. Temporal modulating the coupling strength between the resonators while maintaining the independence of the resonators is crucially important for the study of topological physics (*Phys. Rev. Lett.* 61, 2015(1988), *Rev. Mod. Phys.* 91, 015006(2019), *Nat. Phys.* 7, 907(2011)). We envisioned the extended research work in topological dynamics and Floquet physics based on the proposed metasurface (see Supplementary Note 13 for details). Moreover, the superconducting microbridge can be replaced by a superconducting Josephson junction, which is a nonlinear element and has macroscopic quantum properties. Thus, this design is probably extended to explore novel nonlinear and quantum behaviors. Hence, our work provides a “new platform” for studying time-varying metasurfaces, as stated by previous Reviewer 1.

-2. *The proposal for a topological device is quite intriguing. However, it should be noted that a concept similar to the other proposed application has already been discussed, though with certain differences in implementation, as detailed in the work of F. Miyamaru et al. (Ultrafast Frequency-Shift Dynamics at Temporal Boundary Induced by Structural-Dispersion Switching of Waveguides, Phys. Rev. Lett. 127, 053902, 2021).*

Reply: Thank the reviewer for the recommendation of the relevant work. We have cited this work and added the relevant description in the third paragraph of the introduction section of the revised manuscript. *“In recent work, efficient frequency conversion, as well as phase coherence of the converted waves, could be observed experimentally by ultrafast modulation of structural dispersion in the waveguide⁴¹ or loss in the Fabry–Perot cavity⁴².”*

We have added the following sentence in the second paragraph of the discussion section in the revised manuscript. *“The linear frequency conversion via a temporal boundary can also be extended to other classical systems such as waveguides⁴¹. We envisioned the application enabling efficient on-chip frequency conversion (see Supplementary Note 14 for details).”*

-3. *In the discussion of efficiency, it would be advantageous for the authors to also address the relative shift of the new frequency content compared to the original. Prior studies suggest that efficiency improves as the relative frequency shift is reduced. Thus, a discussion on efficiency that does not include specifics about the frequency shift may not adequately reflect the underlying dynamics.*

Reply: Thank the reviewer for the constructive comment. The relationship between the conversion efficiency and the relative frequency shift is a valuable topic. Since the resonance frequency of the two identical meta-atoms is as high as 0.48 THz, we did not observe the frequency conversion at the frequency of high-order modes. Because the

resonant frequency of meta-atoms without coupling is not tunable, it is difficult to study the relationship between conversion efficiency and frequency shift for different-order modes in experiments as in the previous work (*Nanophotonics* 11, 2045 (2022)).

Fig. R2 Calculated conversion efficiency (η) versus frequency shift (Δf) under different t_{pp} when $|J|_{\max} = 0.167$ THz.

In our work, the frequency shift of the converted wave (Δf) and the conversion efficiency (η) can be adjusted by changing the time delay (t_{pp}). We first analyze the relationship between Δf and η under different t_{pp} when $|J|_{\max} = 0.167$ THz. The relationship of calculated η versus Δf is shown in Fig. R2. Here, $\Delta f = f_c - f_{in}$, where f_c is the peak frequency of the converted wave, and f_{in} is the center frequency of the input wave. The value of η first increases with the increasing Δf , and reaches a maximum of 6.4% at a Δf of 0.135 THz. Then, η gradually decreases as Δf increases. When Δf is above 0.18 THz, η drops to 0. The experimentally measured curve of η versus Δf has a similar trend. The above results are different from previous works (*Nat. Photonics* 12, 765 (2018), *Nanophotonics* 11, 2045 (2022)), which demonstrates η increases monotonously with the increase of Δf . It may be due to the fact that η is affected by both t_{pp} and Δf in our work.

Based on the proposed theoretical model, we also investigate the relationship between Δf and η by altering the coupling strength ($|J|_{\max}$). Δf mainly depends on the frequency difference of the resonance modes when the two resonators are in strong coupling and weak coupling state. Since the frequency splitting in the strong coupling

state depends on $|J|_{\max}$, we can tune Δf by changing $|J|_{\max}$.

Fig.R3 Calculated transmission spectra when $|J|_{\max} = 0.30$ (a) and 0.11 THz (b), respectively. (c) Calculated power spectra of the input pulse with a center frequency of 0.17 THz and the output wave when $|J|_{\max} = 0.30$ and $t_{pp} = -3.8$ ps. (d) Calculated power spectra of the input pulse with a center frequency of 0.37 THz and the output wave when $|J|_{\max} = 0.11$ and $t_{pp} = -3.8$ ps.

In our calculation, we adjust $|J|_{\max}$ between 0.10 and 0.30 and calculate the frequency conversion effect. **Figure R3a, b** show the calculated transmission spectra when $|J|_{\max} = 0.30$ and 0.11 , respectively. The frequency splitting between the two resonance dips of f_1 and f_2 when $|J|_{\max} = 0.30$ is remarkably larger than that when $|J|_{\max} = 0.11$. **Figure R3c, d** show the calculated power spectra of the input and output wave when $|J|_{\max} = 0.30$ and 0.11 , respectively. It shows that the frequency conversion component is much higher when $|J|_{\max} = 0.30$.

The calculated η versus Δf under different $|J|_{\max}$ when $t_{pp} = -3.8$ ps, is shown in

Fig.R4. When $|J|_{\max}$ is below 0.16, η decreases sharply with the decrease of $|J|_{\max}$ while the decrease of Δf is very limited. When $|J|_{\max}$ is higher than 0.16, η increases, and Δf increases with increasing $|J|_{\max}$. Based on our calculation results, we did not see a trend demonstrated in previous works (*Nat. Photonics* 12, 765 (2018), *Nanophotonics* 11, 2045 (2022)) that η increases monotonously with the increase of Δf . It may be because the resonance modes before and after the temporal boundary both have higher Q factors, and the value of η mainly depends on the contrast of the transmission spectra through the temporal boundary. The discrepancy also reflects that the working mechanism of our proposed device is different from previous work.

Fig.R4 Calculated η versus Δf under different $|J|_{\max}$ when $t_{pp} = -3.8$ ps.

To fully address the valuable comment, we have added the calculation results and the above discussion in Supplementary Note 15. Please refer to the revised supplementary information.

-4. The manuscript's claim to be the first in demonstrating 'coherent' frequency conversion via a temporal boundary is misleading and inaccurate. This aspect, particularly the importance of relative phases, has been comprehensively addressed in earlier work, indicating that a revision or further clarification in the manuscript might be necessary.

Reply: Thank the reviewer for pointing out the inaccuracy in our statement and drawing

our attention to the earlier works on "coherent" frequency conversion via a temporal boundary. We regret that some pioneering works in the phase controllability of the converted waves were overlooked in the previous manuscript. We have revised the introduction to describe the earlier theoretical and experimental works on coherent frequency conversion. (*Nat. Photonics* 12, 765 (2018), *Nanophotonics* 11, 2045 (2022), *Phys. Rev. Lett.* 127, 053902 (2021)).

Our innovation in this work is experimentally demonstrating a planar metasurface that achieves highly efficient and phase-controllable frequency conversion via introducing a temporal boundary. A significant difference between our work and the earlier published works is the realization of the ultrafast switching of coupling strength.

To address this nice comment fully, we have made the following revisions:

(1) We have amended the title in the revised manuscript as follows: "*Linear and phase controllable terahertz frequency conversion via ultrafast breaking the bond of a meta-molecule.*"

(2) We have amended the following sentence of the abstract section in the revised manuscript. "*The frequency and relative phase of the converted wave exhibit strong dependence on the pump-probe delay, indicating phase controllable wave conversion.*"

(3) We have amended the following sentences in the third paragraph of the introduction section of the revised manuscript. "*Linear THz frequency conversion was experimentally observed in such metasurface platforms^{39,40}. It was theoretically proposed that phase control can be used for wavefront engineering of the converted waves. In recent work, efficient frequency conversion, as well as phase coherence of the converted waves, has been observed experimentally by ultrafast modulation of structural dispersion in the waveguide⁴¹ or loss in the Fabry–Perot cavity⁴². Despite the progress, a chip-compatible planar frequency converter with a high conversion efficiency and phase control is highly desired to advance the practical applications.*"

(4) We have amended the following sentences in the fourth paragraph of the introduction section of the revised manuscript. "*Phase-controlled linear frequency conversion represents a promising approach for developing new THz sources, which*

are highly sought after in applications such as next-generation communication, imaging^{43, 44}, and radio astronomy⁴⁵.”

(5) We have amended the following sentences in the fifth paragraph of the introduction section in the revised manuscript. *“In this study, we present a conceptual demonstration of linear and phase controllable THz frequency conversion with high efficiency by ultrafast “breaking” the bond of the superconducting-metal hybrid meta-molecule.”*

(6) We have amended the title after Fig. 2 in the revised manuscript. *“Phase controllable frequency conversion in time-varying meta-molecules.”*

(7) We have amended the following sentences in the second paragraph after Fig.3 in the revised manuscript. *“The prediction of the theoretical model is in good agreement with the experimental results, strongly confirming the phase controllable frequency conversion.”*

(8) We have amended the following sentence in the second paragraph of the discussion section in the revised manuscript. *“The efficient achievement of linear frequency conversion is attributed to the time-varying bond strength between meta-atoms induced by the THz pump.”*

-5. Furthermore, a recent study that utilized a temporal boundary in a Fabry-Perot cavity to enhance conversion efficiency, achieving up to 33% (significantly higher than what is reported in this manuscript), should be considered. This study (K. Lee et al., Resonance-enhanced spectral funneling in Fabry-Perot resonators with a temporal boundary mirror, Nanophotonics, 2022) achieved a frequency shift of approximately ~0.2 THz from an original peak frequency of around 0.5 THz, with the newly generated frequencies distinctly separated from the input spectrum. This work is pertinent to the claims made in the manuscript and deserves attention.

Reply: Thank the reviewer for recommending the highly relevant paper. In this work, the authors investigated the spectral and energy conversion efficiency in two scenarios:

single-cycle THz pulses and multi-cycle THz pulse inputs. In the case of a single-cycle pulse, the peak frequency of the input spectrum converges to the resonant frequency of the Fabry-Perot (FP) cavity, leading to enhanced spectral components at that frequency due to a sudden increase in the Q-factor from 4.8 to 48. The energy conversion efficiency reaches 33%, indicating a significant enhancement at the resonant frequency, which is a promising outcome. In the second scenario, similar to our study, when the center frequency of the input signal is 0.6 THz, the energy conversion efficiencies for the fundamental (0.42 THz) and second harmonic (0.8 THz) are 2.6% and 4.3%, respectively. In our work, the planar resonator structure was adopted. Despite the lower Q-factor of the planar resonator compared with the FP cavity, the efficiency values obtained in our work are comparable to this study.

This work has two important reference implications for us. First, the phase controllability of the converted wave is demonstrated experimentally. Second, the high Q factor of the resonator is beneficial to enhance the efficiency of frequency conversion. It is consistent with our results in our electrical tuning experiments (see Supplementary Note 10 for more details), which found that the conversion efficiency decreases as the loss increases.

To fully address this nice comment, we have cited this work and added the relevant description in the third paragraph of the introduction section of the revised manuscript. *“In recent work, efficient frequency conversion, as well as phase coherence of the converted waves, has been observed experimentally by ultrafast modulation of structural dispersion in the waveguide⁴¹ and loss in the Fabry–Perot cavity⁴².”*

We have added a sentence in the fourth paragraph after Fig.3 in the revised manuscript. *“In recent work, the highly efficient linear frequency conversion at the temporal boundary was experimentally demonstrated, which was attributed to the high Q factor of the resonant cavity⁴².”*

In conclusion, I believe that addressing these points with appropriate depth and context would significantly enhance the manuscript. Without these clarifications, it would be

difficult for me to recommend this work for publication in Nature Communications.

Reply: We sincerely thank the reviewer for valuable comments. We have addressed all the comments raised by the reviewer in the replies above. In summary, our work presents a new platform for THz frequency conversion with a quite different operating principle. We believe we have made substantial progress in improving the conversion efficiency and phase controllability in a planar structure both experimentally and theoretically. Our method could be extended to explore other types of time-varying devices to study novel physical phenomena, such as topological photonics and non-Hermitian physics. Our planar design facilitates its integration with waveguide structures on a chip, contributing to the application of on-chip THz frequency converters.

Reviewer 3

The authors have made significant effort to address all the reviewers' comments and have provided corresponding changes to the article main text and the supplementary material. Especially, very good agreement between experiment in model is now achieved with Figure 3d, which shows the phase relation between the input and converted signal. The scientific quality of the newer version is thus strongly improved, and I have no further objectives to publications.

Reply: We are very grateful to the referee for the positive comments on our revised version. It recognizes the efforts we have put into amending the manuscript. We thank the reviewer again for valuable comments that improved the scientific quality of this paper.

Reviewer #2 (Remarks to the Author):

In their rebuttal letter, the authors have attempted to address each raised issue and comment. Their efforts have resulted in a revised manuscript that, in my view, no longer presents any significant concerns. Given the quality of the current version, I am pleased to recommend its publication in Nature Communications.